# An assessment of the teacher completed 'Early Years Foundation Stage Profile' as a routine measure of child developmental health

Kate E. Mooney[1,2]*, Charlie Welch[1], Gareth Palliser[1], Rachael W. Cheung[1,2], Dea Nielsen[1,2], Lucy H. Eddy[3], Sarah L. Blower[1,2]

1 Department of Health Sciences, University of York, York, United Kingdom, 2 Better Start Bradford Innovation Hub, Bradford Institute for Health Research, Bradford, United Kingdom, 3 Department of Psychology, University of Bradford, Bradford, United Kingdom

* kate.mooney@york.ac.uk

## Abstract

The routine measurement of children's developmental health varies across educational settings and systems. The Early Years Foundation Stage Profile (EYFSP) is a routinely recorded measure of a child's development completed at the end of their first school year, for all children attending school in England and Wales. Despite widespread use for research and educational purposes, the measurement properties are unknown. This study examined the internal consistency and structural validity of the EYFSP, investigating whether the summed item-level scores, which we refer to as the 'total score', can be used as a summary of children's developmental health. It also examined predictive validity of the total score with respect to later academic attainment and behavioural, social, and emotional difficulties. The data source was the longitudinal prospective birth cohort, Born in Bradford (BiB), and routine education data were obtained from Local Authorities. The internal consistency and structural validity of the EYFSP total score were investigated using Confirmatory Factor Analysis and a Rasch model. Predictive validity was assessed using linear mixed effects models for Key Stage 2 (Maths, Reading, Grammar/ Punctuation/Spelling), and behavioural, social, and emotional difficulties (Strengths and Difficulties Questionnaire). We found that the EYFSP items demonstrated internal consistency, however, an Item Response model suggested weak structural validity (n = 10,589). Mixed effects regression found the EYFSP total score to predict later academic outcomes (n = 2711), and behavioural, social, and emotional difficulties (n = 984). This study has revealed that whilst caution should be applied for measurement of children with close to 'average' ability levels using the EYFSP, the EYFSP total score is an internally consistent measure with predictive validity.

**Data availability statement:** Data cannot be shared publicly as they are available through a system of managed open access. Researchers interested in accessing the data can find details and procedures on the Born in Bradford website (https://borninbradford.nhs.uk/research/how-to-access-data/). Data access is subject to review by the Born in Bradford Executive, who review proposals on a monthly basis. Requests can be submitted to borninbradford@bthft.nhs.uk. Data sharing agreements are established between the researcher and the Bradford Institute for Health Research. The full analysis code for the IRT analyses is publicly available at https://osf.io/s6num/.

**Funding:** This work was supported by the National Lottery Community Fund (previously the Big Lottery Fund) as part of the A Better Start programme (Ref 10094849). The funder was not involved in the design of the study and collection, analysis, and interpretation of data nor in writing the manuscript. SLB is supported by the NIHR Yorkshire and Humber Applied Research Collaboration (ARC-YH; Ref: NIHR200166, see https://www.arc-yh.nihr.ac.uk,). The views in this publication are those of the authors and not necessarily those of the NIHR or the Department of Health and Social Care. LHE was supported by an ESRC Postdoctoral Fellowship (ES/X006050/1).

**Competing interests:** The authors have declared that no competing interests exist.

## Introduction

'Developmental health' is a broad concept that combines a holistic understanding of physical, mental, social, and emotional wellbeing, combined with core educational abilities such as mathematics and literacy [1]. Measurements of children's early developmental health can be used to predict later educational performance and health [2–4], which are both, in turn, important predictors of adult social and health outcomes [5,6]. Ensuring that children have strong developmental health in the earliest years of their lives can therefore contribute to their future educational attainment [7] and, consequently, help to close socioeconomic inequalities in educational outcomes [8,9]. It is therefore important to routinely measure children's developmental health using an accurate and valid measure to identify those who may need extra support [10,11].

At an international level, widely used measures of child developmental health include the Early Childhood Development Index (ECDI) and the Early Development Instrument (EDI) [12]. The ECDI has been in use since 1990 by the United Nations as an indicator of 'improved access to quality early childhood development, care and pre-primary education', and consists of 10 caregiver-reported questions for children aged 3–5 years across literacy, numeracy, learning/cognition, physical and socioemotional development [13].

Some countries have utilised existing educational settings to monitor children's developmental health with teacher-based assessments, such as the EDI and the Teaching Standard GOLD® (TS GOLD) measure. The EDI has been widely implemented across Canada (all provinces except for one since 1998) and Australia (all schools since 2009) [14], and assesses children aged 4–6 years-old, regarding their physical health, wellbeing, social competence, emotional maturity, language and cognitive development, communication skills, and general knowledge across 103 items [15,16]. The EDI has generally demonstrated adequate psychometric properties in terms of internal consistency (with all domains except for physical health having Cronbach's alpha > 0.86). However, it has demonstrated variable model fit (with CFI and TLI values all above .80 for all subscales in Canada, Australia, and the USA, and RMSEA values ranging between 0.063 and 0.228) [17], and variable predictive validity (with Pearson's r correlations ranging between 0.19 and 0.38 between the Language and Cognitive Development domain scores and the Peabody Picture Vocabulary Test, PPVT) [17].

In the US, the Teaching Standard GOLD® (TS GOLD) teacher-report measure has been used as a formative assessment of developmental health in children aged 2–5 years in 19 out of 54 public preschool programmes as of 2012, and is also used by federal Head Start programmes as of 2019 [18]. Teachers rate children across social-emotional, physical, language, cognitive, literacy, and mathematics domains from Level 0 ('Not Yet') to Level 9 ('Exceeds kindergarten expectations'). TS GOLD has demonstrated adequate measurement of each domain as a latent construct (factor loadings ranged between 0.68 – 0.95), but variable model fit (SRMR ranging between 0.38 to 0.50; CFI between 0.90 to 0.92; RMSEA between 0.06 to 0.07). It has demonstrated metric, scalar, and strict invariance across longitudinal measurements, good interrater agreement between teachers and experienced raters (all above 0.80), and poor to moderate concurrent validity with the Bracken School Readiness Scale (ranging ICC between 0.38 to 0.54; Pearson's r for individual scale scores between 0.27 to 0.74) [19].

Overall, although teacher-based assessments have the advantage of using existing educational settings to assess children's developmental health, providing much-needed population data on child development [14], as well as limiting the stress children may experience from formalised exam-based assessments [20], further research is needed to understand the psychometric strengths and limitations of teacher-based assessments as general developmental health measures.

Within England and Wales, the Early Years Foundation Stage (EYFS) was introduced in 2008 to provide a research-based framework with information on how children learn and

develop, aimed at practitioners to assist them in delivering high quality early years environments [21]. Based on the EYFS framework, the EYFS 'Profile' (EYFSP) was introduced as a teacher assessment of children's development and learning, completed at the end of the academic year in which the child turns five [22]. It was originally introduced with 69 'Early Learning Goals' (ELGs). Following a review which indicated a need to simplify and reduce the number of goals for teachers to complete [21], a new profile consisting of 17 ELGs was introduced in 2012. Whilst specific, detailed information regarding how the specific ELGs were chosen is limited, and the EYFSP was not developed as a robust measurement tool (in comparison to, for instance, the EDI), the ELGs do appear to relate to children's early developmental health. The ELGs span seven different developmental areas; 'Communication and language development', 'Physical development', 'Personal, social and emotional development', 'Literacy', 'Mathematics', 'Understanding the world', and 'Expressive arts and design' [23,24].

In the version of the EYFSP that we analyse in this study (second version, delivered 2012-2021), the EYFSP is scored according to whether a child meets each ELG as "Emerging", "Expected" or "Exceeding". A revised version of the EYFSP has been available since 2021, with changes to the content and focus of the ELGs, and children are scored as only "Emerging" or "Expected" [25] (see Supporting Information File 1 for second and revised versions). The present study investigates the second version of the EYFSP as this was used nationally and routinely for nine years, and cohort studies have utilised it in research studies, both as an outcome in evaluations of interventions or policies [26], and as a predictor in association studies [27]. Despite the update to the revised version, data from second version is likely to continue to be relevant in the future, as there are several studies listed on the ISRCTN that are using the EYFSP as an outcome, and protocols for evaluations which plan to use it as an outcome in the future [28].

## The 'Good Level of Development' (GLD)

The EYFSP has been predominantly used in research studies and educational monitoring as a binary measure, where children either meet a 'Good Level of Development' (GLD), or they do not. Children are scored as having achieved a GLD if they have achieved *at least* the expected level for the ELGs in the core areas of "communication and language", "physical development", "personal, social and emotional development", "literacy" and "mathematics" [23]. The Department for Education monitors national and regional averages of children reaching a GLD and compares the number of children achieving GLD across different groups according to characteristics such as gender and eligibility for free school meals [29].

Further, several research studies have investigated risk factors for not achieving a GLD. Children with 'English as an Additional Language' (EAL) status have been found to have lower proportions of GLD achievement in comparison to native English-speaking children [30], and children born later in the academic year are much less likely to achieve a GLD [31–33]. Additionally, children achieving the GLD have higher odds of performing at expected levels on later academic assessments at age 7 [34], and lower odds of later being identified as having Special Educational Needs or Disability (SEND) [35].

Whilst the GLD is a useful benchmark to establish which children are meeting the core components of the EYFSP, it has important limitations. Dichotomising variables (continuous or categorical) is problematic for two key reasons. First, much information is lost, so the statistical power to detect an association using the variable is reduced substantially [36]. In fact, dichotomising a variable can reduce statistical power by the same amount as would discarding a third of the data or more [37]. Second, dichotomisation can lead to an underestimation of the extent of variation in outcome between groups, as individuals close to but on opposite sides of the cut point are characterised as being very different rather than very similar [36].

Applying the GLD method to the EYFSP therefore means missing out potentially valuable information on the number of goals for which a child meets or exceeds. It means that children very close to, but on opposite sides of, the GLD threshold are characterised as being very different, despite meeting or exceeding a similar number of goals. For instance, children who meet zero goals, and children who meet eleven out of twelve GLD goals, would be scored as '0' on the GLD. The GLD also essentially ignores the distinction between children who are "Expected" and "Exceeding" in various goals, as a child who scores "Expected" in all the GLD goals, and a child who scores "Exceeding" in all the GLD goals would both be scored as a 1. As children vary considerably across different developmental areas during early childhood [38,39], this simple GLD approach is a very limited assessment of children's developmental health. In summary, much of the variation in the EYFSP items, and thus the variation in developmental health amongst children, is ignored by the GLD measure.

## The 'total score'

An alternative to the GLD is to instead assign numerical scores to each category in the EYFSP (e.g., 0 for emerging, 1 for expected, and 2 for exceeding in the revised version; or 0 for emerging and 1 for expected in the newer version), and sum these scores into a 'total score' (resulting in a score ranging between 0–34 for the original version, and 0–17 for the revised version). This approach overcomes the above limitations that are found with using the GLD, as it better captures the variation in EYFSP responses.

Nonetheless, the EYFSP total score has been seldom used in research studies in comparison to the GLD. Previous research has considered the impact of early years workforce qualifications on children's later EYFSP total scores [40,41]. One study found the original version of the EYFSP to be predictive of later language, literacy, and mathematics [42]. Since then, only one study has used the revised version of the EYFSP total score to predict later outcomes, finding it to be a strong predictor of later Autism Spectrum Disorder diagnoses for children within the Born in Bradford cohort [43]. Importantly, there are no studies exploring the psychometric measurement properties of the EYFSP total score.

## Subscale scores of the EYFSP

As described earlier, there are seven individual learning areas within the EYFSP. However, associations between the seven individual areas of the EYFSP and later related outcomes have not been extensively explored. This may provide information about the predictive validity (i.e., the extent to which a test is an adequate reflection of a 'gold standard') of the specific areas [44]. For instance, do the 'personal, social and emotional development' areas have significant predictive associations with a validated measure of children's social and emotional development? If so, this specific area (with a score ranging between 0-12) could be used as an outcome in isolation, meaning that intervention studies aiming to improve children's social and emotional development could use this area with the three goals as an outcome. This rationale can be generalised to all seven areas of the EYFSP.

The preliminary evidence on whether the individual areas significantly relate to other outcomes is promising, but very limited. Children with higher language comprehension scores achieved higher scores on the EYFSP writing scale, however, the writing scale is no longer in the current version of the EYFSP [45]. In the Born in Bradford cohort, EYFSP scores relating to literacy and physical development were found to predict total difficulties on the Strengths and Difficulties Questionnaire (SDQ) [27]. However, the EYFSP scores relating to literacy and physical development are not the most relevant subscales for the

SDQ total difficulties score, and it was not reported how the EYFSP subscale scores were calculated for this particular study.

The EYFSP is recommended for use in educational settings to assess children's strengths and weaknesses, and whether they need support in a particular area [23]. Information about the predictive validity of the specific areas will provide confidence in doing this, as it will validate whether the areas are predictive of later outcomes.

## Rationale and objectives

The EYFSP total score has huge potential to provide useful information on children's early developmental health that could be utilised for research and educational purposes, at both a population and individual level. Despite the EYFSP being administered to over 7.5million children since being introduced [34], there is an absence of any psychometric research on it. Specifically, there is no previous research on the internal consistency or structural validity of the EYFSP 'total score', nor any research on its predictive validity for academic outcomes. Research is therefore needed to establish whether the EYFSP 'total score' is fit for purpose in both research studies and applied educational settings.

We first investigate the structural validity of the EYFSP total score; that is, the degree to which the total score reflects the dimensionality of the construct to be measured [44]. We achieve this using Item Response Theory (IRT); a set of psychometric models for developing and refining psychological measures [46]. To accompany this, we investigate the internal consistency of the EYFSP; that is, the degree of the interrelatedness among the items which represents the extent to which all items of a test measure the same construct [44,47].

We also investigate the predictive validity of the EYFSP total score, to assess the degree to which it predicts future outcomes [44]. Since it is assumed that measures administered at the start of school can provide an understanding into children's future attainment, establishing predictive validity is crucial [4]. Whilst the predictive validity of the EYFSP GLD has been investigated [34,35], the predictive validity of the EYFSP total score for academic outcomes has not been investigated. We investigate whether the EYFSP total score is predictive of children's later academic outcomes at age 10–11 years, and investigate whether specific EYFSP subscales (relating to communication and socioemotional wellbeing), are predictive of children's behavioural, social, and emotional difficulties.

In summary, we had five aims that assessed two key aspects of using the EYFSP for research and educational purposes:

**Internal Consistency/Structural validity of the EYFSP:**

1) Investigate whether the EYFSP items demonstrate internal consistency

2) Investigate whether the EYFSP items demonstrate structural validity, i.e., that the total scores from the instrument can be used as a summary measurement that represents children's early school skills

Predictive Validity of the EYFSP:

3) Investigate if the EYFSP total score predicts children's later academic attainment (for maths, reading, and grammar/punctuation/spelling)

4) Investigate if the EYFSP total score predicts children's later behavioural, social, and emotional difficulties

5) Investigate if the EYFSP subscales (relating only to communication and socioemotional wellbeing) predict later behavioural, social, and emotional difficulties

## Materials and methods

### Design

This study comprises secondary data analyses of an observational birth cohort in Bradford, England.

### Setting

The data source is the longitudinal cohort study, Born in Bradford (BiB). The BiB cohort recruited pregnant mothers between March 2007 and December 2010 at the Bradford Royal Infirmary. All babies born to these mothers were eligible to participate and more than 80% of women invited agreed to participate [48]. The cohort comprises of 12,453 mothers, 13,776 pregnancies and 3,448 fathers. At recruitment, the two largest ethnic groups in the sample were Pakistani heritage (45%) and White British (40%), followed by Indian (4%) and Asian Other (3%) [49].

Mothers completed the BiB baseline questionnaire when they were recruited and reported information on family demographics and socioeconomic indicators. Routine education data relating to personal characteristics and educational outcomes were obtained from the Local Authority every year that the child attends school, starting at age 4 (Reception year). Additional bespoke data were collected by Born in Bradford on children aged 7 to 10 years in 89 Bradford schools between 2016 and 2019, including a teacher reported Strengths and Difficulties Questionnaire (SDQ) (which is the outcome for Research Questions 4-5) [38]. Born in Bradford and the 'Primary School Years' wave received ethical approval for the data collection from the NHS Health Research Authority's Yorkshire and the Humber—Bradford Leeds Research Ethics committee (references: 07/H1302/112, 16/YH/0062). Informed written consent was obtained for all parents recruited.

### Internal consistency and structural validity analyses

The analyses were preregistered at osf.io/s6num. Data were combined and cleaned using Stata/MP 18.0. Internal validity analyses were completed using the *mirt* and [50,51] *ggmirt* [52] packages in R.

**Measurements.** The EYFSP total score was summed from the 17 Early Learning Goals (ELGs) in the profile.

As seen in Table 1, each area of learning contains specific goals. The EYFSP handbook provides a description of each goal and what a child must achieve to meet each level [24]. Practitioners are instructed to review the evidence gathered in order to make a judgement for each child and for each ELG, and then to score each ELG as either:

- *Emerging:* not yet at the level of development expected at the end of the EYFS

- *Expected:* best described by the level of development expected at the end of the EYFS

- *Exceeding:* beyond the level of development expected at the end of the EYFS

The EYFSP handbook instructs that practitioners must make their final EYFSP assessments based on all their evidence, where 'evidence' means any "material, knowledge of the child, anecdotal incident or result of observation, or information from additional sources that supports the overall picture of a child's development" [24].

The responses to each ELG and how they were coded in this study are as follows: 'Emerging' = 0, 'Expected' = 1, and 'Exceeding' = 2. If children were absent from school for a long period of time, this is marked on their records and these children were dropped from the

**Table 1. Overview of the ELG's within the EYFSP and the area of learning they relate to.**

| Area of Learning | EYFSP ELGs (See Section 6 of 2020 EYFSP handbook for further detail [24]) |
|---|---|
| Communication and language development | 1. Listening and attention*<br>2. Understanding*<br>3. Speaking* |
| Physical development | 4. Moving and handling*<br>5. Health and self-care* |
| Personal, social and emotional development | 6. Self-confidence and self-awareness*<br>7. Managing feeling and behaviour*<br>8. Making relationships* |
| Literacy | 9. Reading*<br>10. Writing* |
| Mathematics | 11. Numbers*<br>12. Shape, space and measures* |
| Understanding the world | 13. People and communities<br>14. The world<br>15. Technology |
| Expressive arts and design | 16. Exploring and using media and materials<br>17. Being imaginative |

Note: Asterisks are the ELGs that a child must achieve at least 'expected' level into achieve a GLD.

analyses. The EYFSP total score was summed from the 17 ELGs (see Table 1), and therefore ranged between 0–34.

**Analysis.** We used Item response theory (IRT) to assess the structural validity of the EYFSP total score [46]. IRT can be used to assess whether creating a total score from the items is appropriate and assess the strength of relationships between items and constructs of interest. Item response models assume the latent trait variable is reflected by a unidimensional continuum (i.e., item responses are explained by one latent continuous variable, or single dimension). We fitted a polytomous 'Rating Scale' version of the 1-parameter logistic Rasch model, since the items have more than two possible response categories (see further details under 'Rasch model parameters') [53]. Under the Rasch model, two test takers who both achieved, for example, 12 EYFSP items, but who achieved a different set of items would receive the same ability estimate [54]. This allows us to interrogate the structural validity of the summed 'total score'.

**Rasch model parameters.** Let $Y_{ij}$ denote the response to item $i$ for child $j$, with $Y_{ij}$ taking the values 0 ('Emerging'), 1 ('Expected') or 2 ("Exceeding"). The polytomous rating scale Rasch model posits that the probability of child $j$ with latent ability $\theta_j$ obtaining responses 0, 1 or 2 for item $i$ are given by:

$$\Pr\left(Y_{ij}=0\,|\,b_i,d_1,d_2,\theta_j\right)=\frac{1}{1+\exp\left[\theta_j-\left(b_i+d_1\right)\right]+\exp\left[\left[\theta_j-\left(b_i+d_1\right)\right]+\left[\theta_j-\left(b_i+d_2\right)\right]\right]}$$

$$\Pr\left(Y_{ij}=1\,|\,b_i,d_1,d_2,\theta_j\right)=\frac{\exp\left[\theta_j-\left(b_i+d_1\right)\right]}{1+\exp\left[\theta_j-\left(b_i+d_1\right)\right]+\exp\left[\left[\theta_j-\left(b_i+d_1\right)\right]+\left[\theta_j-\left(b_i+d_2\right)\right]\right]}$$

$$\Pr\left(Y_{ij}=2\,|\,b_i,d_1,d_2,\theta_j\right)=\frac{e^{\left[\left[\theta_j-\left(b_i+d_1\right)\right]+\left[\theta_j-\left(b_i+d_2\right)\right]\right]}}{1+\exp\left[\theta_j-\left(b_i+d_1\right)\right]+\exp\left[\left[\theta_j-\left(b_i+d_1\right)\right]+\left[\theta_j-\left(b_i+d_2\right)\right]\right]}$$

where $b_i$. denotes the overall difficulty of item $i$ and $d_1$, $d_2$ denote the distances between adjacent response categories (common across all items). Furthermore, it is assumed that $\theta_j \sim N(0, \sigma_\theta^2)$ and that the item discrimination parameters are 1 across all items. This contrasts with conventional Rasch parameterisation which constrains the item discrimination parameters to be constant across all items (but not equal to unity) and assumes the latent ability $\theta_j$ to be distributed $N(0, 1)$.

The item difficulty parameter measures the difficulty of achieving a higher scoring response, whereas the discrimination parameter is a measure of the differential capability of an item (i.e., a high discrimination value suggests an item that has a high ability to differentiate between subjects with similar, latent abilities) [55]. In a Rasch model, discrimination is constrained to be equal across all items, and difficulty is estimated separately for all items [54]. The polytomous rating scale version of the Rasch model also includes category threshold parameters which are constrained to be equal across items, and provide a measure of the distances between the difficulties of adjacent levels of response for each item.

**Model fit.** The fit of the Rasch model was assessed using Root Mean Square Error of Approximation (RMSEA), where values of $< 0.02$ with sample sizes of $1000+$ indicate that the data do not underfit the model [56]. We also report the Comparative Fit Index (CFI) (values $> .90$ are acceptable), and Standardised Root Mean Square Residual (SRMR) (values $< .08$ are acceptable) [57].

**Item fit.** Item infit and outfit indicate how well the item responses fit the model [58]. Item fit was assessed using infit/outfit statistics, with values between 0.5 and 1.5 considered to be acceptable [59] and RMSEA as described above.

**Local dependence.** Local dependence is the assumption that the only influence on an individual's item response is that of the latent trait variable being measured and that no other variable (e.g., other items on the EYFSP scale) is influencing individual item responses. We used the 'residuals' function in the *mirt* package to examine the standardised local dependency $\chi^2$ statistic (where any correlation higher than the average item residual $+ .2$ [60] classifies as local dependency).

**Item Response Theory visuals.** The test information function shows a measure of the information provided by the total test score across the range of latent ability levels (denoted $\theta$). Information is a statistical concept that refers to the ability of a test (or item) to reliably measure the latent ability $\theta$. The test characteristic curve shows the relationship between the total summed score on the y axis, and latent ability ($\theta$) on the x axis [61]. Plots of item characteristic curves and item information functions are provided at osf.io/s6num/.

**Unidimensionality.** We tested unidimensionality with a Confirmatory Factor Analysis (CFA) of a latent trait with all EYFSP items loading onto it and examined McDonald's hierarchical Omega, which reflects the percentage of variance in the scale score accounted for by a single general factor. This allows us to estimate the extent of internal consistency among the EYFSP.

## Predictive validity analyses

The analysis plan for the predictive validity analyses was preregistered at osf.io/s6num. Following pre-registration, we made two changes to the analytic plan. These were: (1) the inclusion of a binary term for Special Educational Needs and Disabilities (SEND) status as a covariate in all analysis models and (2) inclusion of 'school at time of outcome' as a random intercept in all analysis models. Inclusion of SEND status as a control covariate was necessary as children with SEND may have lower EYFSP scores relative to typically developing children [62]. Inclusion of 'school at time of outcome' as a random intercept was necessary as EYFSP scores may vary across schools. All analyses for this component of the research were undertaken using Stata/MP 18.0.

**Measurements.** There were two separate predictors for this analysis. For the measurement of EYFSP total score, see the above section.

For the measurement of EYFSP Communication and Socioemotional goals (EYFSP-CS), we tested the strength of the association between the 'communication and language' and 'personal, social, and emotional' ELG's with children's outcomes. This EYFSP-CS score ranged between 0-12 and was obtained by summing the responses to the six items in the two relevant areas.

There were two separate outcomes for this analysis. For Research Question 3, we measured Academic attainment via the Key Stage 2 Assessment completed towards the end of Year 6 at school by children when aged 10-11. In educational records, there are separate continuous scaled scores for (1) Maths, (2) Reading, and (3) Grammar/Punctuation/Spelling that range between 0 and 120. Any children who scored '0' were excluded from the analyses, as any children with '0's recorded are pupils who have achieved too few marks to be awarded a scaled score [63]. Analysed scores therefore ranged between 80 and 120.

For Research Questions 4-5, we used the Strengths and Difficulties Questionnaire (SDQ) to measure children's behavioural, social, and emotional difficulties [64]. The SDQ was collected once for children when they were aged 7-10 in the 'Primary School Years' wave. The 25 items in the SDQ comprise five scales of five items each. 'Somewhat True' is always scored as 1, but the scoring of 'Not True' and 'Certainly True' varies with the item. A total difficulties score is generated by summing scores from all the scales (emotional symptoms, conduct problems, hyperactivity, peer relationships) except the prosocial scale, and the resultant score ranges from 0 to 40, where a higher score indicates higher difficulties

Table 2 below provides an overview of all covariates included in both models. Covariates were included in the regression models if they were thought to be confounders of the association between EYFSP and the outcome, or if they were covariates that would be expected to improve the precision of our estimates.

**Analysis models.** All research questions were answered using linear mixed effects models, with fixed effects of socioeconomic status, parent immigration status, child ethnicity, SEND, child age, and child language as covariates (see Table 2), and a random intercept for school at the time of outcome measurement. The four outcomes were: (1) Reading, (2) Maths, (3) Grammar, Punctuation, and Spelling, and (4) SDQ. The SDQ scores were analysed twice, once using EYFSP total score as a predictor, and once using EYFS-CS subscale as a predictor. The model for each outcome can be described as;

$$\delta_{ij} = \beta_0 + \beta_1 EYFSP\,score_{ij} + \beta_2 Child\,EAL_{ij} + \beta_{3\&4} Child\,ethnicity_{ij}$$
$$+ \beta_5 Parent\,immigration\,status_{ij} + \beta_{6\&7\&8\&9} Socioeconomic\,status_{ij}$$
$$+ \beta_{10} SEND_{ij} + \beta_{11} Child\,Age_{ij} + U_j + \varepsilon_{ij}$$

Where $\delta$ is each outcome, $\beta_0$ is the intercept, each $\beta$ is a coefficient, $u_j$ is the random intercept for school $j$, and $\varepsilon_{ij}$ is the residual error for individual $i$ within school $j$. The letters identify the levels within the model, where $i$ is the individual and $j$ is the school. Child ethnicity$_{ij}$ & Socioeconomic status$_{ij}$ represent a set of dummy variables.

Unstandardized regression coefficients and Wald method 95% confidence intervals based on variance estimates obtained via Rubin's rules are reported for all models [70].

**Missing data methods.** We used Multiple Imputation using Chained Equations (MICE) to impute missing data on parent immigration status, socioeconomic position, and SEND (see Fig 1 for numbers of missing values), under the assumption that the missing values are missing at random (MAR). Briefly, data are MAR if the probability of the data being missing does not depend on the unobserved measurements/values, conditional on the observed data

**Table 2. Overview of covariates in all models.**

| Variable (with evidence for relationship to exposure/outcome) | Variable type (scale) | Details |
|---|---|---|
| EYFSP score (exposure) | Continuous (0–34) | Modelled via a single linear term |
| Child English as an Additional Language (EAL) (confounder) [65] | Binary (0/1) | Coded as 0 = English is first language, 1 = English is an Additional Language |
| Child ethnicity (confounder) [66] | Categorical (0/1/2) | Coded as 0 = White British, 1 = Pakistani, 2 = Other |
| Parent immigration status (confounder) [67] | Binary (0/1) | Coded as 0 = Born in UK, 1 = Born outside of UK |
| Socioeconomic status (confounder) [8,68] | Categorical (0/1/2/3/4) | "Most economically deprived" = 0, "benefits and not materially deprived = 1, "employed and no access to money = 2, "employed and not materially deprived" = 3, "Least socioeconomically deprived and most educated" = 4.<br>Derived from a previously validated measure of socioeconomic position in Born in Bradford [69]. See Supporting Information File 2 (Attachment A) for the characteristics of the socioeconomic groups. |
| Special educational needs and/or disability (SEND) (confounder) [68] | Binary (0/1) | Coded as 0 = No SEND, and 1 = Any SEND (including children with an EHCP). |
| Child age at time of outcome (covariate) [68] | Continuous | Child age in months is recorded for Research Question 1, and child age in years is recorded for Research Question 2 (due to data availability). Both modelled via a single linear term in the respective analyses |
| School at time of outcome (multi-level variable) | Categorical | Modelled via a random intercept |

[70,71]. While the validity of this approach to analysis rests on assumptions about the nature of the missing data, and indeed the appropriateness of the imputation and substantive analysis models, we believe that these assumptions serve as reasonable approximations to reality in the present context, and are certainly more plausible than the assumptions underpinning the analysis that excludes the incomplete cases.

Every variable that was in the analysis model was included in the imputation model. Eligibility for free school meals (binary, no missing values) was also included in the imputation model. We used Stata's 'mi impute chained' command to generate 25 imputed datasets for each research question. The results section presents the pooled results from the multiply imputed datasets (results from analyses based on complete cases were similar).

**Robustness checks.** Model fit was assessed between models run with (1) EYFSP modelled as a continuous variable via a single linear term and (2) EYFSP as an unordered categorical variable modelled via a series of dummy variables. Model fit assessed via AIC and BIC was marginally better with EYFSP as a continuous variable, and the continuous modelling provides a more parsimonious estimate, so this model was selected. A scatter plot of fitted and residual values was considered to show no evidence of heteroskedasticity.

**Effect sizes.** Half of a standard deviation has been previously found to correspond to a minimum clinically important difference [72,73]. We therefore calculated half of a standard deviation in the outcomes, and compared these to our effect estimates. The outcomes, standard deviations, and effect sizes of interest are provided in Table 3.

## Results

### Participants

Fig 1 shows the total number of recruited BiB children (n = 13,858), and the numbers within each measurement and analyses set.

**Table 3. Standard deviations and effect sizes of interest for all outcomes.**

|  | Standard deviation | Effect size of interest (unstandardised) |
|---|---|---|
| Maths | 7.05 | 3.52 |
| Reading | 8.16 | 4.08 |
| Grammar/Punctuation/Spelling. | 8.09 | 4.05 |
| Behavioural, social, and emotional difficulties (SDQ) | 6.26 | 3.13 |

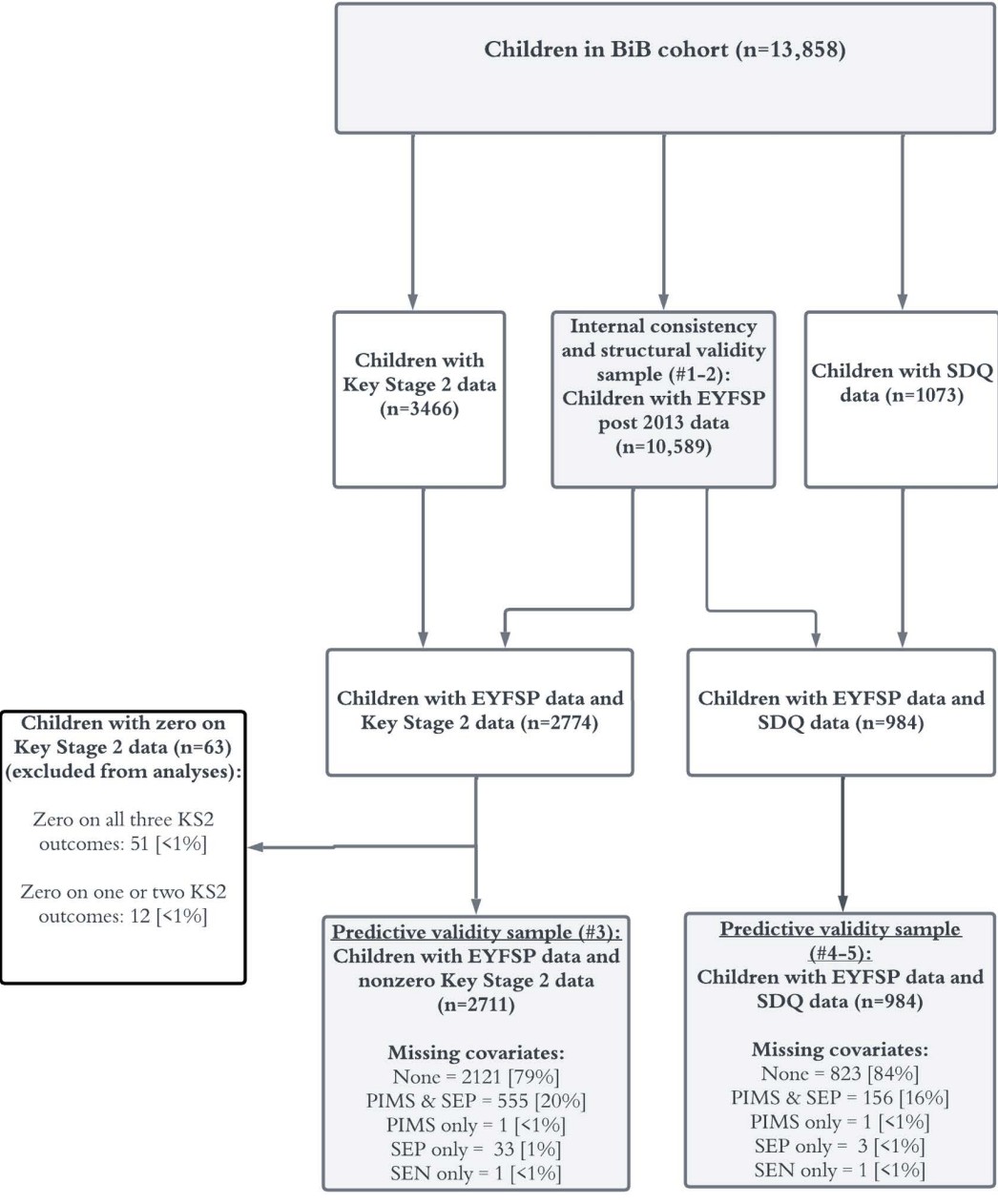

**Fig 1. Flow chart of included study participants..**

## Descriptive information

Table 4 describes the sample for all children who had EYFSP data (n = 10,589). The mean EYFSP total score in this sample was 15.30 (SD = 8.07), and it ranged between 0-34. The mean EYFSP score within children who achieved a GLD (n = 6,272, 59%) was 20.38 (SD = 4.96), and scores ranged between 12-34. The mean EYFSP score within children who did not achieve a GLD (n = 4,317, 41%) was 7.92 (SD = 5.67), and scores ranged between 0-27.

Fig 2 further demonstrates that there is considerable overlap in total scores between children who do and do not achieve a GLD. It also demonstrates that there is substantial variability in scores within children who do and do not achieve a GLD.

## Item response theory analysis

Full analyses with the code, results, and additional sensitivity analyses are provided at https://osf.io/s6num/.

**Structural validity: Rasch model parameters, model fit, and item fit.** The model fit values (RMSEA = 0.138, SRMSR = 0.162, CFI = 0.938) indicated poor fit to the overall Rasch model. The maximum likelihood estimates of the category threshold parameters were -3.585 and 3.473 and the maximum likelihood estimate of the variance of the latent ability was 9.532

**Table 4. Descriptive information on sample for all children with complete EYFSP data (n = 10,589).**

| Variable | N (%) |
|---|---|
| **Ethnicity** | |
| White British | 3,650 (34%) |
| Pakistani | 4,874 (46%) |
| Other[*] | 2,063 (19%) |
| Missing | 2 (<1%) |
| **Socioeconomic Position[**]** | |
| Most deprived | 1,414 (13%) |
| Benefits but coping | 2,649 (25%) |
| Employed no access to money | 1,354 (13%) |
| Employed not materially dep | 1,730 (16%) |
| Least deprived and most educated | 1,529 (14%) |
| Missing | 1,913 (18%) |
| **Parent immigration status** | |
| Parent born inside UK | 5,551 (52%) |
| Parent born outside UK | 3,168 (30%) |
| Missing | 1,870 (17%) |
| **English as an Additional Language** | |
| Yes | 4,662 (44%) |
| No | 5,753 (54%) |
| Missing | 174 (2%) |
| **Special Educational Needs** | |
| No | 8,345 (79%) |
| Yes | 2,132 (20%) |
| Missing | 112 (1%) |

[*]The most populous ethnic groups within other were Indian (16% of the Other group, Bangladeshi (14%), Other Asian (14%), and Other White (14%).

[**]socioeconomic groups listed in Supporting Information File 2: Attachment A and in Fairley et al (2014).

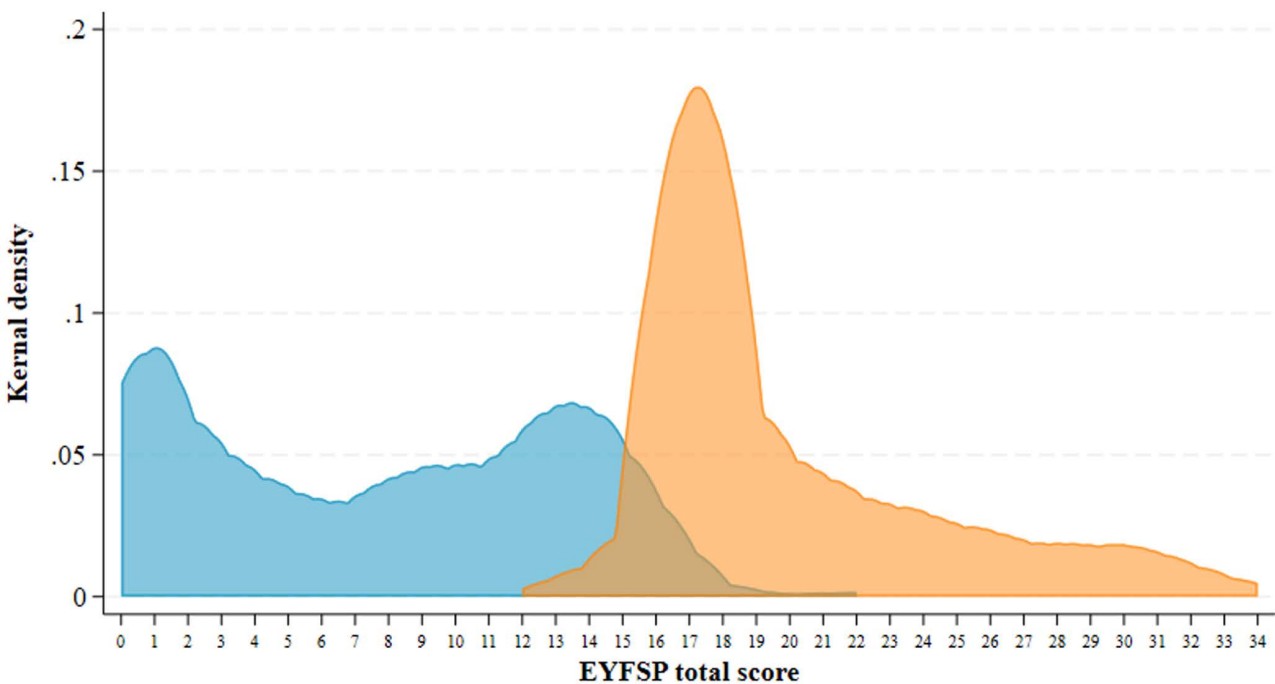

**Fig 2. Kernel density distributions of EYFSP total score for those who do not achieve a GLD (in blue) and do achieve a GLD (in orange) (n = 10,589).**

(discrimination parameter constrained to be equal to 1). We next assessed the item parameters and the item fit values for the overall Rasch model.

Table 5 shows that the easiest item is 'Moving and handling' (goal 4), and hardest item is 'Writing' (goal 10). The item fit values show that Item 9 has the highest RMSEA value (although other items also have problems with misfit). The item infit/outfit values are provided at osf.io/s6num/ and generally indicated values within the acceptable range.

**Local dependence.** The local dependency matrix is presented in osf.io/s6num/. The matrix identifies a local dependence issue between Items 2 & 3 (communication items) (residual = .44); and Item 9 & Item 10 (the literacy items) (residual = .48).

**Test information function and test characteristic curve.** Fig 3 demonstrates that most information is provided at the lower/higher ends of ability (i.e., those children with latent abilities at least one standard deviation above/below the mean latent ability). It also shows that less information is provided for children with close to average abilities - shown by the dip in the curve around $\theta = 0$. Fig 4 presents the scale characteristic curve, showing the relationship between the total summed score on the y axis, and the overall latent ability ($\theta$) on the x axis. The test shows good discrimination for children with latent abilities that are slightly-to-moderately higher and lower than average (i.e., $\theta \in [-5, -1] \cup [1,5]$), and slightly less powerful discrimination for children with close to average abilities (shown by the flattening in the curve around $\theta = 0$), and for children with very high or low abilities (shown by the flattening of the curve at the more extreme values of $\theta$.

**Internal consistency.** The CFA indicated high factor loadings (all > .8) onto one construct, and a parallel analysis indicated that a one factor model was a reasonable representation of the data [74]. We assessed internal consistency using McDonald's hierarchical omega, finding a point estimate of 0.89.

**Table 5. Rating Scale Model parameters and item fit values.**

| Item | Diffi-culty | θ such that Pr(Emerging\|θ) = Pr(Expected\|θ) | θ such that Pr(Expected\|θ) = Pr(Exceeding\|θ) | RMSEA | P value |
|---|---|---|---|---|---|
| 1. Communication and language: Listening and attention | 0.000 | −3.585 | 3.473 | 0.040 | <.001 |
| 2. Communication and language: Understanding | 0.037 | −3.548 | 3.510 | 0.050 | <.001 |
| 3. Communication and language: Speaking | 0.446 | −3.139 | 3.919 | 0.034 | <.001 |
| 4. Physical development: Moving and handling | −0.147 | −3.732 | 3.326 | 0.038 | <.001 |
| 5. Physical development: Health and self-care | −0.050 | −3.635 | 3.423 | 0.035 | <.001 |
| 6. Personal, social and emotional: Self-confidence and self-awareness | 0.125 | −3.460 | 3.598 | 0.022 | <.001 |
| 7. Personal, social and emotional: Managing feelings and behaviour | 0.317 | −3.268 | 3.790 | 0.025 | <.001 |
| 8. Personal, social and emotional: Making relationships | 0.146 | −3.439 | 3.619 | 0.032 | <.001 |
| 9. Literacy: Reading | 1.206 | −2.379 | 4.679 | 0.070 | <.001 |
| 10. Literacy: Writing | 1.975 | −1.610 | 5.448 | 0.050 | <.001 |
| 11. Mathematics: Numbers | 1.431 | −2.154 | 4.904 | 0.041 | <.001 |
| 12. Mathematics: Shapes, space and measures | 1.408 | −2.177 | 4.881 | 0.032 | <.001 |
| 13. Understanding the world: People and communities | 1.224 | −2.361 | 4.697 | 0.025 | <.001 |
| 14. Understanding the world: The world | 1.311 | −2.274 | 4.784 | 0.022 | <.001 |
| 15. Understanding the world: Technology | 0.607 | −2.978 | 4.080 | 0.063 | <.001 |
| 16. Expressive arts and design: Exploring and using media and materials | 0.869 | −2.716 | 4.342 | 0.034 | <.001 |
| 17. Expressive arts and design: Being imaginative | 1.110 | −2.475 | 4.583 | 0.030 | <.001 |

*Note: Column two shows the estimated item level difficulty, where higher values indicate greater difficulty. Columns three and four show the values obtained by adding the estimated category threshold parameters (-3.585 and 3.473) to the estimated difficulty parameters. These show the values of the latent ability θ such that the probabilities of a participant achieving adjacent responses are equal.*

## Predictive validity analysis

**Academic attainment outcome.** The mean scores and standard deviations for the Key Stage 2 outcomes were Maths = 105.08 (7.05), Reading = 103.67 (8.16) and Grammar/Punctuation/Spelling (GPS) = 106.64 (8.09). For full regression results and the key analyses code, please see Technical Appendix File 2 (Attachment C). All models indicated that a higher EYFSP total score was associated with a higher Key Stage 2 outcome. Key results are described and displayed below.

*For the Maths outcome (n = 2711),* the model explained a significant amount of the variance (unadjusted $R^2$ = .33; $F_{(11,23939.7)}$ = 124.65, $p < .001$). Higher EYFSP total scores were associated with higher Maths scores (B = 0.356 [0.322 to 0.390], $p < .001$).

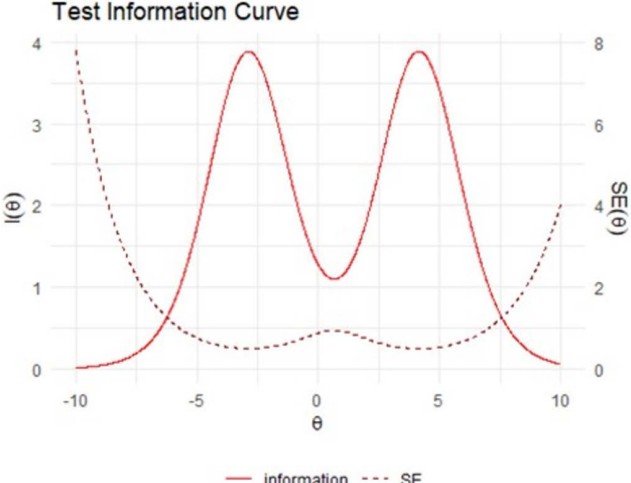

**Fig 3. Test Information Curve for θ∈ [-10, 10].**

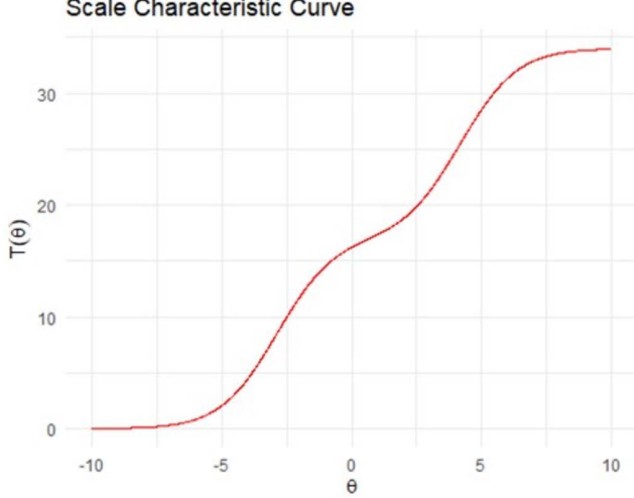

**Fig 4. Scale characteristic curves representing total score (0-34) across ability (θ∈ [-10, 10]) based on the fitted model.**

For the Reading outcome *(n = 2711)*, the model explained a significant amount of the variance (unadjusted $R^2 = .31$; $F(11,22414.3)) = 108.91$, $p < .001$). Higher EYFSP total scores were associated with higher Reading scores (B = 0.424 [0.384 to 0.464], $p < .001$).

For the GPS outcome *(n = 2711)*, the model explained a significant amount of the variance (unadjusted $R^2 = .37$; $F(11,18477.5) = 146.05$, $p < .001$). Higher EYFSP total scores were associated with higher GPS scores (B = 0.427 [0.390 to 0.464], $p < .001$).

Fig 5 displays the association between a difference in EYFSP goals (ranging between 1-10), and the estimated change in outcome in the different academic outcomes. For instance, an increment of 1 EYFSP total score point results in a change of between 0.36 to 0.42 in the outcomes, and an increment of 10 results in a change of between 3.56 to 4.24. To reach a minimum clinically important difference (shaded area in Fig 3), the difference in EYFSP total

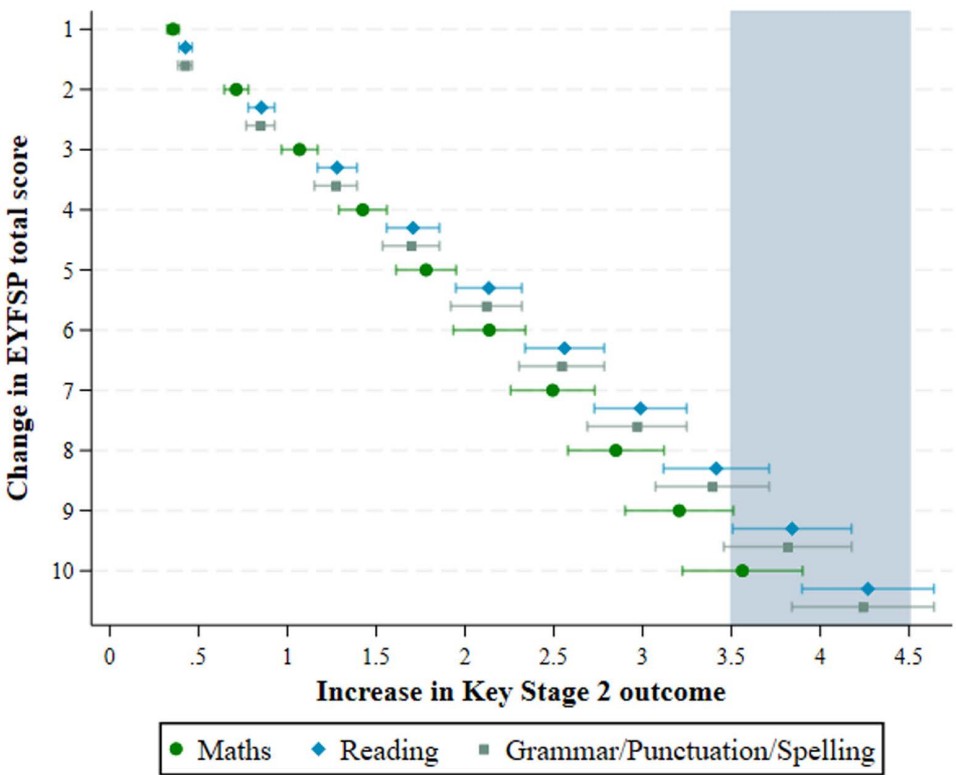

**Fig 5. Increase in 'EYFSP total score' and 'EYFSP-CS score' associated with change in Academic Outcomes (Maths, Reading, Grammar/Punctuation/Spelling).**

score required is approximately 8 for Reading and Grammar/Punctuation/Spelling, and 10 for Maths.

**Behavioural, social, and emotional difficulties outcome.** The mean EYFSP-CS score was 5.78 (SD = 3.16). The mean SDQ score was 7.31 (SD = 6.26). For full regression results, please see Technical Appendix File 2, Attachment D. Key results are described and displayed below. Note that a higher score on SDQ indicates more socioemotional difficulties.

When we included the EYFSP total score as the predictor (n = 984), the model explained a significant amount of the variance (unadjusted $R^2$ = .25, $p < .001$; F(11,66324.9) = 27.04). The EYFSP total score was associated with a decrease in the SDQ total difficulties (B = -0.20 [-0.26 to -0.15], $p < .001$).

When we included only the EYFSP-CS predictor (n = 984), the model explained a significant amount of the variance (unadjusted $R^2$ = .25, $p < .001$; F(11,67390.5) = 26.72). The EYFSP-CS sores were associated with a stronger decrease in the SDQ total difficulties (B = -0.48 [-0.61 to -0.37], $p < .001$).

Fig 4 displays the association between an increase in EYFSP goals (ranging between 1–10), and the estimated change in outcome for the socioemotional wellbeing measure (SDQ). For instance, a change of 1 in the EYFSP total score results in a change of -.20 in SDQ, and a change of 1 in EYFSP-CS results in a change of -.48 in SDQ. A change of 6 in the EYFSP total score results in a change of -1.22 in SDQ, and a change of 6 in EYFSP-CS results in a change of -2.90 in SDQ (with the confidence interval crossing over the clinically important difference) (Fig 6).

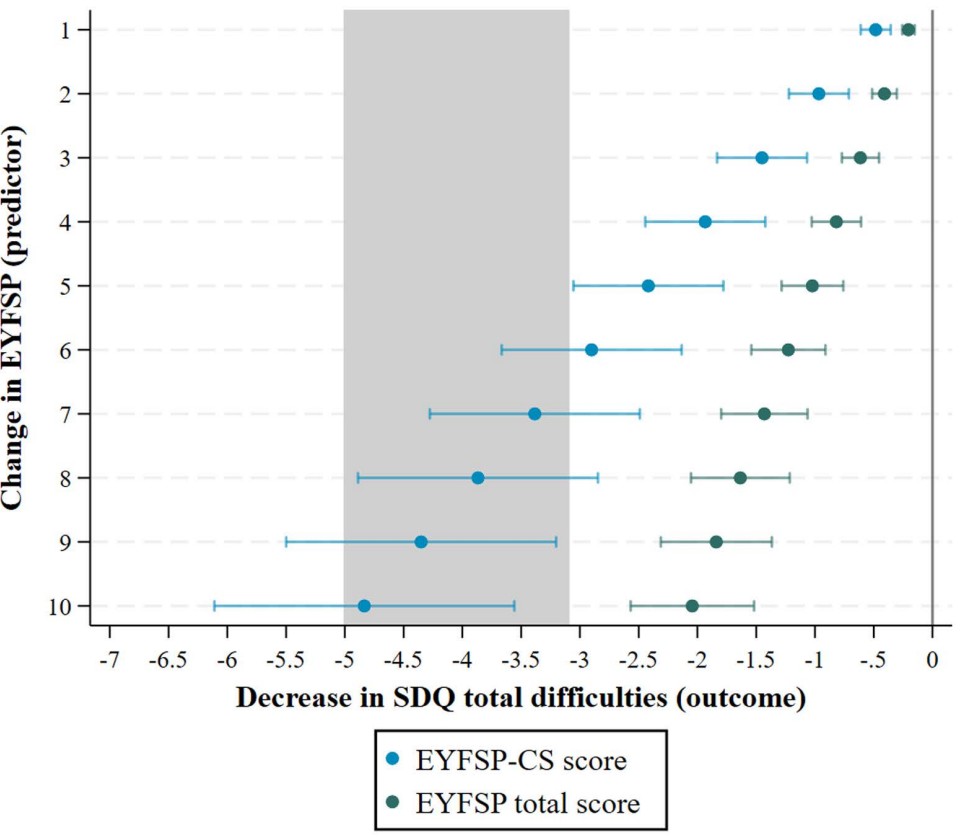

**Fig 6. Increase in 'EYFSP total score' and 'EYFSP-CS score' associated with change in SDQ total difficulties score.** Note: estimates were produced using Stata user written command xlincom. The shaded area represents an estimated minimum clinically important difference.

## Discussion

The first aim of this study was to investigate the internal consistency the EYFSP items [44,47]. The EYFSP items demonstrated high internal consistency, with results indicating that the items primarily measure one unidimensional construct. We tentatively suggest that the measured construct is compatible with the definition of children's 'developmental health'. The construct of developmental health encompasses a holistic understanding of children's physical, mental, social, and emotional wellbeing, combined with core educational abilities such as mathematics and literacy [1]. This reflects the EYFSP's original purpose to operate as a research-based framework of children's learning and development [21,75].

The second aim was to investigate if the EYFSP demonstrated structural validity. The IRT analyses indicated a poor fit to the polytomous Rasch model. However, the test information and scale characteristic curves show the total score provides substantial information across a wide range of underlying ability, with some loss of precision at very close to average abilities. This indicates that whilst the test provides information across a wide range of ability levels, it provides relatively less information for children with 'average' latent abilities (e.g., the 40% of children between roughly the 35th and 75th percentiles). This means that two children with equal scores of, for example, 16, may have different ability levels in reality (e.g., one could have slightly below average ability and one slightly above), but that the EYFSP total score is not able

to precisely discriminate between them. It should also be noted that an IRT Rasch model is extremely restrictive, as it requires all items to be equally discriminating, and this is very rarely the case in measurements of person ability [54].

Although IRT has been used to examine measures of, for example, emotional dysregulation [76] and neuropsychological capabilities [77] of young children, we have not been able to locate any previous studies using IRT on a comparable measure of broad child developmental health. The closest study is an investigation of the 'Denver Developmental Screening Test (DDST)', though the aim of this study was to apply IRT to develop a scoring method to estimate 'ability age' of individual children using the DDST [78]. Hence, IRT has been underused in the context of examining the structural validity of child developmental health measures. Our study is therefore first to investigate the EYFSP's internal consistency and structural validity using IRT, and one of the first to apply this method to early child developmental health.

Nonetheless, the internal consistency and structural validity of comparable child developmental health measures has been investigated using other psychometric analysis methods, namely the EDI and TS GOLD. The EYFSP has now demonstrated similar adequate internal consistency as the EDI [17], and demonstrates model fit similar to both the EDI and TS GOLD, which have also shown poor model fit [17,19]. Model fit refers to the ability of a model to reproduce the data, with poor model fit indicating that relationships between variables may be incorrectly specified in the applied statistical model [57]. Given that the EYFSP, EDI, and TS-GOLD have all demonstrated poor model fit, this may indicate a general challenge with using teacher reported measures of overall child developmental health. As this is such a broad construct, perhaps it is challenging to assess in one holistic measure. Indeed, a systematic review of parent-reported measures of child social and emotional wellbeing/behavior found such measures to have structural validity [79].

The poor model fit may be explained by misspecification of individual items. Item misfit can arise due to multidimensionality (where the item relates to a separate latent trait) and/or poor item quality [58]. The items with worst fit were two of the 'Literacy' area items, and one of the 'Understanding the World' items. This may indicate that these items do not measure the latent trait of children's developmental health, and that their removal may improve the model fit. However, Literacy is one of the core areas of the EYFSP and is crucial for teachers to be able to assess. Hence, this item could instead be replaced with another, similarly worded item that reflects 'Literacy', but has a better fit to the model.

The poor model fit may also be due to the less precise estimates of ability evident for children with 'average' ability, which may relate to the varying administration of the measure in educational settings [40,41]. The administration is not standardised or moderated, and therefore susceptible to considerable variation. Additionally, the procedures and requirements of the EYFSP may not lend themselves to identification of more nuanced differences in ability for children with generally average levels of development. The high number of children meeting expected levels of development in all 17 goals is potentially indicative of this issue. More guidance for teachers on how to identify differences in children's abilities, as well as more robust procedures for moderating scores, could potentially address the apparent issues with reduced precision for children with close to average abilities, and increase the information provided by the measure.

Evidence from comparable measures of child developmental health comes from numerous psychometric studies conducted in several different countries over several years [17,19]. Hence, to be able to thoroughly compare the EYFSP to these similar measures, more research on the measurement properties of the EYFSP is needed (see implications and future directions section).

Our third aim was to investigate the predictive validity of the EYFSP total score for academic outcomes. We found that the EYFSP total score strongly and consistently predicts

academic outcomes at ages 10-11 in Maths, Reading, and Grammar, Punctuation and Spelling assessments. It has been previously found that the EYFSP GLD is predictive of children's academic outcomes at ages 6-7 during Key Stage 1 [34], and the present study extends this finding to the EYFSP total score, and to Key Stage 2 assessments at ages 10-11 years. To reach an important change in academic outcomes (considered to be half the standard deviations of the observed Key Stage 2 scores), a difference in EYFSP total score of 8-10 points was required (dependent on the outcome). This information will be useful for researchers to note if they wish to use the EYFSP total score as an outcome for intervention studies. For instance, as a difference in EYFSP total score of 8 was required to reach an important difference for the Reading outcome, and this could be used as a benchmark for future educational interventions which aim to improve children's reading abilities. Though, the estimates reported in Fig 3 could also be used to identify differences in the EYFSP total score that translate to smaller differences in these outcomes, which may serve as more realistic target differences for future intervention studies.

Our fourth and fifth aims were to explore the predictive validity of the EYFSP total score and the EYFSP-CS subscales for children's behavioural, social, and emotional difficulties at ages 7-10. The relevant EYFSP-CS subscales had a much stronger association with behavioural, social, and emotional difficulties than the EYFSP total score. A difference of 6 points for the EYFSP-CS score was associated with important differences in behavioural, social, and emotional difficulties, whereas no changes in the EYFSP total score were associated with important differences. Again, a difference of 6 (for the EYFSP-CS score) could be used as a benchmark for future interventions which aim to improve children's behavioural, social, and emotional abilities (or translated for a more realistic target difference). Researchers can more confidently use the communication and social subscales to measure behavioural, social, and emotional difficulties.

There is only one other study that has reported predictive validity of this version of the EYFSP subscales; it found that EYFSP scores relating to literacy and physical development also predicted children's behavioral, social, and emotional difficulties [27]. In comparison to other measures of child developmental health, the EDI has demonstrated variable predictive validity between the language and cognitive development domain scores and the Peabody Picture Vocabulary Test [17]. The TS GOLD has been found to be associated with children's assessments throughout the school year [80], and has been found to have variable concurrent validity with the Bracken School Readiness Scale [19]. The EYFSP therefore has demonstrated adequate predictive validity in comparison to other measures of child developmental health, however, there are substantially fewer studies regarding the EYFSP.

## Implications and future directions

Although this study has highlighted some limitations of using the EYFSP, we do not suggest the use of a different measure of child developmental health over the EYFSP. Whilst other measures of child developmental health have undergone further psychometric analyses, they too demonstrate some variable psychometric properties (e.g., variable model fit values have been demonstrated in both the EDI [17] and TS-GOLD [19]). Replacement of the EYFSP would be a significant overhaul to current educational practice and should be avoided if possible. Hence, once the measurement properties of the EYFSP are investigated as described below, the EYFSP could be used with more confidence than it currently is. Or, if the measurement properties of the EYFSP are found to be significantly lacking, a significant programme of development should be undertaken to develop it further, or replace it with an already validated instrument of child developmental health.

However, the findings from this study do support future use of the EYFSP total score over the EYFSP GLD score for research and educational purposes. Although both the GLD and now

the total score have been shown to predict future outcomes [34,35], there is substantial variation in total scores within children who do and do not achieve a GLD (see Fig 1). The GLD therefore does not capture the variation in children's developmental health that the EYFSP total score does, and it has no other evidence regarding its measurement properties. We therefore recommend that researchers use the EYFSP total score instead of the GLD, if it suits their study purpose. For teachers, the GLD is a useful metric for identification of children who may later be diagnosed with special educational needs [35], however, teachers may wish to also examine a child's EYFSP total score to gain a more nuanced understanding of a pupils' development. Though, it is important to note that the EYFSP total score should be used with some caution when making inferences about 'average' ability children (those with total scores between approximately 15 and 18).

There is still much to be learnt about the measurement properties of the EYFSP. It would be beneficial to directly compare the measurement properties of the EYFSP total score to the GLD in a future study, explicitly examining whether more valid and accurate conclusions can be made about child developmental health using the EYFSP total score than can be done using the GLD. There are also other measurement properties which could be tested, including the content validity (the degree to which the EYFSP reflects children's developmental health as a construct) and criterion validity (the degree to which the EYFSP items and summaries derived thereof are an adequate reflection of a gold standard). This would require collection of an additional measure of child development at the same time of the EYFSP, perhaps the EDI, since this is implemented in a comparable way at population level, and has undergone substantial development since its inception [15,81]. In terms of predictive validity, we explored associations between EYFSP-CS and the SDQ, due to the Born in Bradford sample facilitating this analysis with a sufficient sample size, providing an ideal test case. However, more research is needed to explore whether each specific goal area is associated with other measurements (e.g., literacy to reading assessments, physical activity to motor skill measurements).

Finally, it will be important to test the measurement invariance of the EYFSP over socio-demographic subgroups. Given the possibility that teacher-based assessment may systematically underassess minority groups [82], we suggest a future study should explore the validity of the measure across various socioeconomic and ethnic groups.

## Limitations

This study specifically considered data from the second version of the EYFSP, which was administered between 2012 to 2021. We tentatively suggest that the findings regarding internal consistency and predictive validity will generalise to the revised version of the EYFSP, as the items (which contain similar wording and measure the same areas, see Technical Appendix File 1) should remain internally consistent with one another and predictive of future outcomes, even with less variation in the data. However, as data from the revised EYFSP becomes widely available, future research will need to test if the findings in this study generalise to the current version. This is particularly important as the structural validity may be significantly affected by the changes to the content of the ELGs, along with the removal of one of the response categories (the 'Exceeding' category). Removal of this category may result in a 'ceiling' effect for populations, with most children scoring 'expected' on all categories. While this change may be acceptable for educational purposes, it may have a negative impact on the usefulness of the EYFSP total score for research purposes.

This study included only children in the Born in Bradford cohort, and therefore may only be relevant for comparable populations with high levels of deprivation and a diverse ethnic population. Whilst the ethnic diversity of this sample improves the generalisability of the findings to ethnically diverse populations, we did not explore the measurement properties of the EYFSP within specific ethnic groups.

## Conclusions

While the EYFSP has been utilized as a measure of children's early developmental health, this was not its intended purpose. Despite this, the present study has revealed that whilst caution should be applied for measurement of children with close to 'average' ability levels, the EYFSP total score is an internally consistent measure with predictive validity. The EYFSP total score also provides better information for children with very high and very low abilities. Given that the EYFSP was not developed as a robust measurement tool, the EYFSP total score appears to be a reasonable measure of child developmental health for routine use in England and Wales.

## Supporting information

**S1 File. Overview of the Early Learning Goals (ELGs) in the old and new versions of the Early Years Foundation Stage Profile (EYFSP).**
(DOCX)

**S2 File. Predictive Validity Analysis.**
(DOCX)

## Acknowledgments

Born in Bradford is only possible because of the enthusiasm and commitment of the children and parents in BiB. We are grateful to all the participants, health professionals, schools and researchers who have made Born in Bradford happen.

## Author contributions

**Conceptualization:** Kate E. Mooney, Lucy H. Eddy, Sarah L. Blower.

**Data curation:** Kate E. Mooney.

**Formal analysis:** Kate E. Mooney, Charlie Welch, Gareth Palliser.

**Investigation:** Kate E. Mooney.

**Methodology:** Kate E. Mooney, Charlie Welch, Gareth Palliser, Rachael W. Cheung, Dea Nielsen, Lucy H. Eddy, Sarah L. Blower.

**Project administration:** Kate E. Mooney.

**Visualization:** Kate E. Mooney, Charlie Welch.

**Writing – original draft:** Kate E. Mooney, Gareth Palliser, Rachael W. Cheung, Dea Nielsen, Lucy H. Eddy, Sarah L. Blower.

**Writing – review & editing:** Kate E Mooney, Charlie Welch, Gareth Palliser, Rachael W. Cheung, Dea Nielsen, Lucy H. Eddy, Sarah L. Blower.

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
