## [Decision Letter · Decision Letter 0]

17 Jul 2024

PONE-D-24-14293An assessment of the teacher completed ‘Early Years Foundation Stage Profile’ as a routine measure of child developmental healthPLOS ONE

Dear Dr. Mooney,

Thank you for submitting your manuscript to PLOS ONE. After careful consideration, we feel that it has merit but does not fully meet PLOS ONE’s publication criteria as it currently stands. Therefore, we invite you to submit a revised version of the manuscript that addresses the points raised during the review process.

Thank you for your patience with the review process!  Please revise and resubmit your manuscript after thoroughly addressing all three of the reviewers comments and feedback. These revisions, specifically from reviewer #1, will strengthen your manuscript. ==============================

We look forward to receiving your revised manuscript.

Kind regards,

Nhu N. Tran

Academic Editor

PLOS ONE

Journal Requirements:

2. Please ensure that you refer to Figure 6 in your text as, if accepted, production will need this reference to link the reader to the figure.

Reviewers' comments:

Reviewer's Responses to Questions

**Comments to the Author**

1. Is the manuscript technically sound, and do the data support the conclusions?

Reviewer #1: No

Reviewer #2: Yes

Reviewer #3: Yes

2. Has the statistical analysis been performed appropriately and rigorously?

Reviewer #1: No

Reviewer #2: Yes

Reviewer #3: I Don't Know

3. Have the authors made all data underlying the findings in their manuscript fully available?

Reviewer #1: Yes

Reviewer #2: Yes

Reviewer #3: No

4. Is the manuscript presented in an intelligible fashion and written in standard English?

Reviewer #1: Yes

Reviewer #2: Yes

Reviewer #3: Yes

5. Review Comments to the Author

Reviewer #1: Thank you for the opportunity to review the manuscript, “An assessment of the teacher completed ‘Early Years Foundation Stage Profile’ as a routine measure of child developmental health”, submitted to PLOS One.

The authors report a secondary data analysis, which attempts to establish the internal consistency and structural validity of the Early Years Foundation Stage Profile (EYFS-P) in one ethnically diverse region in England (i.e., Bradford).

In principle, this study has the potential to inform research using this specific administrative data. However, several substantial issues within the current manuscript prevent its publication in its current form and underscore my ‘reject’ recommendation. These issues are outlined in detail below. Overall, I hope this feedback is helpful for future revisions of this work.

‘Developmental health’ definition and implications for structural validity evaluation

Throughout the manuscript, the authors refer to the EYFS-P as a measure of ‘developmental health’ which they define as “a broad concept that combines a holistic understanding of physical, mental, social and emotional wellbeing”.

However, this definition is not well-aligned with the intended purpose of the EYFS-P, which also incorporates “specific areas of learning” (i.e., early academic skills). For example, as highlighted in the DfE (2024) guidance, the EYFS-P is designed to assess the “prime areas of learning (which are: communication and language; personal, social and emotional development; and physical development) and the specific areas of mathematics and literacy” (as well as Understanding the world and Expressive arts and design) (see page 6 https://assets.publishing.service.gov.uk/media/65253bc12548ca000dddf050/EYFSP_2024_handbook.pdf). Moreover, the early learning goals (ELGs) within the EYFS-P are reflective of the early years foundation stage curriculum (see Development Matters documentation) and thus may be better thought of as a curriculum-based measure.

As a consequence of this misalignment, the authors have misattributed the constructs measured within the EYFS-P. This has important implications for their structural validity evaluation. Specifically, they have used a confirmatory factor analysis approach with the a-priori assumption that the EYFS-P loads onto one factor (‘developmental health’).

However, given the design and intended purpose of the EYFS-P, there may be alternative factor structures that should also be considered, for example, a 2-factor structure (prime areas of learning; specific areas of learning) or a 7-factor structure (Communication and language; Personal, social and emotional development; Physical development; Literacy; Mathematics; Understanding the world; Expressive arts and design). As such, an exploratory factor analysis approach would be more appropriate in the current study.

Application of the current findings to current educational practice

Although the authors used the 2008-2012 version of the EYFS-P as the basis for their study, they make the following conclusions: “We expect that the results from this study will generalise to the revised version of the EYFSP, particularly the findings regarding internal consistency and predictive validity” and “This study supports future use of the EYFSP total score over the EYFSP GLD score for research and educational purposes”.

These conclusions are not supported and are problematic because the 2008-2012 version of the EYFS-P is significantly different from current educational practice. For example, the most recent statutory EYFS-P introduced in Sept 2021 includes several substantial changes to the ELGs. These changes include the removal of the Shape, Space, and Measure ELG, for the introduction of the Numerical Patterns ELG, as well as distinct changes to the content of a range of the ELGs across the EYFS-P. These changes will have important implications for the psychometric properties that the authors have examined and thus limit the generalisability of the current findings.

Furthermore, it is of concern that the authors do not acknowledge anywhere in their manuscript that the EYFS-P was revised again in 2021 (e.g., on page 4).

The authors should, therefore, clarify that their results are only applicable to research studies that are planning to use the 2008-2012 EYFS-P total score (page 27). Any further application (without further investigation) would be inappropriate.

Fine-grained analytical approach

Although the authors helpfully adopt a more fine-grained approach for considering the relationships between EYFS-P Communication and Socioemotional goals and SDQ difficulties (page 25), there presents a missed opportunity for adopting a similar approach for literacy-based ELGs to later literacy outcomes, and maths-based ELGs to later maths outcomes. This is despite the authors’ argument on page 6, which states “this rationale can be generalised to all seven areas of the EYFS-P”.

Psychometric terminology

Pages 6-7- The authors write “information about the construct validity (i.e. the extent to which a test measures what it is intended to measure)… or instance, do the ‘personal, social and emotional development’ areas have significant predictive associations with a validated measure of children’s social and emotional development?” This is incorrect, as it is an example of (predictive) criterion validity, not construct validity. I recommend that the authors familiarise themselves with resources, such as the AERA, APA, NCME Standards for Educational and Psychological Testing, which will help ensure that the terminology they use is accurate.

Page 8- The authors write “this [internal consistency] is an essential first step prior to investigating the structural validity of the EYFS-P” – this is not strictly true. Rather the COSMIN taxonomy and the AERA, APA, NCME Standards for Educational and Psychological Testing resources (mentioned above) suggest that structural validity comes first (i.e., how many factors are there?), followed by internal consistency (i.e., are the items interrelated?). Furthermore, the interrelations between items should be considered at both the full-item level (i.e., all EYFS-P items) and (if applicable) at the specific factor level (e.g., if the structural validity evaluation identified 2-factors, then the internal consistency also needs to be reported for these 2 factors). As such, I recommend that the reporting of the results in the current study be re-structured (and re-analysed per the suggestions mentioned).

Other specific issues within the manuscript

Page 3- The authors write “The embedding of standardised measurement of this into educational systems varies greatly across countries”, but then go on to discuss one example (Canada) – are there other examples that could be included here?

Page 3- The authors write “Due to the educational pressures that standardised exam settings can bring, assessments completed by children’s teachers can instead offer a valuable insight” – this is somewhat true, but requires further clarification- e.g., Educational pressures are also not the only reason why teacher-based assessments may be considered optimal with very young children. Likewise, teacher-based assessments can also be ‘high-stakes’, such as those completed in Key Stage 1.

Page 3- The authors write “The EDI… has generally demonstrated adequate psychometric properties in terms of internal consistency (15), and predictive validity”. It would be helpful to report exact figures here, as the term ‘adequate’ is often inconsistently and/or misused across psychometric studies. The COSMIN taxonomy and the AERA, APA, NCME Standards for Educational and Psychological Testing, will provide helpful starting points for the authors in establishing the consistent use of the term ‘adequate psychometric properties’.

Page 4- The authors write “69 ‘early learning goals’ (ELGs)”- this is not strictly true, rather the DfE refer to the structure of the 2008-2012 version of the EYFS-P as “13 assessment scales, with 9 points within each scale (‘scale point’). The 13 assessment scales are grouped into six areas of learning” (e.g., https://assets.publishing.service.gov.uk/media/5a7b875c40f0b62826a0429c/sfr28-2010.pdf and https://dera.ioe.ac.uk/id/eprint/8221/13/Early_FS_Handbook_v11_WO_LR_Redacted.pdf ).

Page 4- The authors write “the original version…has been used nationally and routinely for nine years” – this is also not strictly true, as the 2008-2012 version of the EYFS-P (the focus of the current study) was used for 4 years. The 2012-2021 revised version for a further 9 years, with the latest iteration made statutory in Sept 2021.

Page 6- The authors highlight research evaluating the measurement properties of the EYFS-P, but do not include other relevant literature, e.g., Snowling et al. 2011 https://files.eric.ed.gov/fulltext/ED526910.pdf

Page 6- The authors highlight “there are seven individual learning areas within the EYFSP” – (this refers to the 2012-2021 version), but on page 4 the authors state “the present study investigates the original version” (which I assume to mean the 2008-2012 version), which would include 13 assessment scales (see above point)- further clarity is required here.

It may also be helpful to create a table (perhaps expanding on table 1) that outlines each of the three versions of the EYFS-P, including which learning areas are measured, how they are measured (i.e., meets, exceeds, etc) and when they were implemented (i.e., years).

Page 7- The authors write “In understanding these strengths and weaknesses, a child could then be provided with support in a particular area”- this argument needs to be further developed, as the EYFS-P in its current scoring structure is still designed and used for this purpose by educational practitioners. How is what the authors are suggesting, any different (and superior) from current practice?

Page 15- The authors write “Analysed scores therefore ranged between 80 and 120”- It is not clear how academic attainment was measured - were maths, reading, and grammar considered separately, or combined? what score was used- raw or scaled? it isn't clear why 80 is the minimum score?

Pages 15-16- Further clarity is required about the SDQ measure- e.g., were children assessed every year between the ages 7-10? Or just once? What are the five scales? Is the prosocial scale included in the five scales?

Page 16 – What is your evidence to justify the inclusion of these covariates? “thought to be confounders” is not evidence-based.

Page 20- There is a relatively large proportion of missing data within the socioeconomic position variable (18%), which is comparable to the size of other response options. Although missing data methods are applied, it would be beneficial to conduct additional robustness checks with other socioeconomic-proxy variables, such as children’s eligibility for free school meals or pupil premium. These administrative data are likely to be available for a larger proportion of children in the linked sample that the authors are already using.

Page 20- Were there missing data for other predictor and outcome variables (described on pages 15-16)? If so, how were they handled? What proportion of data were missing?

Page 30- The authors helpfully highlight the ethnic diversity of their sample. However, the authors should expand their discussions to consider any potential SES/ethnic inequalities in teacher-based assessment (rather than child-direct measures).

Page 31- The conclusion “total score is an internally consistent measure with predictive validity” should be further clarified/toned down, as it is only for the very low and very high-performing children; the 'average' child is not adequately assessed. Although the authors later add “we caution against using it for measurement of children with very close to ‘average’ ability levels” – the first highlighted sentence is misleading for readers.

Minor points

Page 3 – “can therefore improve” would be better phrased as “can therefore help contribute to” – or something similar. Currently, the text implies a definitive causal relationship, which excludes the possibility of other factors contributing to later, positive child outcomes.

Page 3- “kindergarten teachers [in Canada]”- what age is this?

Page 8- “future outcomes” – which ones?

Page 10- “an observational birth cohort” – additional text is required to clarify the data is from one region in England (Bradford).

Page 10- “for all women recruited” – however the authors also mention fathers in paragraph 2, page 10. Please clarify the text accordingly.

Page 15- “two key changes to this upon starting the analysis” – what was the rationale for these changes?

Page 15 (and throughout the manuscript)- inconsistent use of SEN/ SEND.

Page 15- make it clear that the “Key Stage 2 assessment” is completed by children, and not a teacher-based assessment.

Page 15- “end of Year 6 at school”- what age is this?

Page 16- use APA guidelines when using numbers in text – i.e., five, instead of 5.

Page 19- “Model fit assessed via AIC and BIC was marginally better with EYFSP as a continuous variable, and the continuous modelling provides a more parsimonious estimate, so this model was selected” – please clarify where these results are reported.

Page 23 (and throughout the manuscript)- the information reported on the OSF page should also be included in a Supplementary materials section in the published manuscript.

Page 24- “EYFSP total score was associated with a higher Key Stage 2 outcome” – This text needs to be clarified – e.g., “higher EYFS-P total scores were…”

Page 24- what does GPS stand for? This is the first time the acronym is used.

Page 25- “Note that a higher score on SDQ indicates more socioemotional difficulties” should be included in the text on page 16 (i.e., SDQ method section).

Page 29- “affected by the removal of one of the response categories” – be clear which one.

Reviewer #2: This manuscript examines the internal consistency and structural and predictive validity of the EYFSP measure using linear mixed effects models, a CFA, and an IRT. This is thorough and timely work, and the statistical tests in the manuscript are appropriate and well-described. The feedback that I have is intended to strengthen the manuscript for publication.

1. In the predictive validity analysis on page 15, please provide a brief justification for the changes made to the pre-registered analysis.

2. Please provide a brief justification for the use of MICE over other missing data imputation methods (i.e., why a MAR method over a MCAR method?).

3. I would like if the implications of the misfit of certain items, as demonstrated by the RMSEA scores and discussed on pages 22-23, were more directly addressed in the discussion. Why might these particular items have misfit issues or be particularly impacted by the issues addressed by the authors on pages 27-28?

4. The authors mention several times that caution should be taken when making inferences about 'average' ability children. I wondered if the authors believe that there is an alternative to the EYFSP total score worth exploring for these children, and what this might be if so.

5. In their limitations, the authors mention that the BiB cohort may not be relevant across other kinds of populations. In what ways do the authors believe that this sample may yield different results as opposed to other samples?

6. Along the same lines as the above, in the methods and materials section, the authors should provide more detail about the demographic makeup of the BiB cohort. They mention in the discussion that this is an ethnically diverse sample. Can specific ethnicity and/or SES demographics be included in the main manuscript beyond the inclusion of the two largest ethnicities represented?

Reviewer #3: The authors take on an extraordinary blind spot in the UK public policy with child development and education at the intersection. Millions of teachers complete millions of measures on millions of children, and never has a psychometric property been considered. The authors clearly list their intentions, and the rationale for the study is well articulated. The authors also thoroughly examine the precedent held where data were compressed into Good Level of Development. This reviewer agrees that the dichotomous representation bleaches the data of nuance.

Little was written in the background as to how the EYFSP was developed, though the brief history of its revision is included. Some of the administration of the EYFSP methods are left to the discussion rather than the background. That the EYFSP is not standardized or moderated leaves the original intent (research) of the tool to collect dust. This reviewer is stunned that a gigantic, incalculable number of person hours is invested in the UK each year on a measure that was never "normed." This is not a criticism of the present manuscript, but rather a reason to accept this manuscript that critiques our well intentions even when they are not evidence-based. I was relieved to learn that there was indeed high internal consistency, but do wonder if the original authors of the measure (or panel, or work group - however it came to pass) developed the EYFSP with some degree of sound methodology.

The authors present novel findings on the extended predictive properties of the total score, now reaching children 10-11. Some inconsistent arguments exist in that the total score is discussed as needing to increase by 8-10 in order to achieve 1/2SD on key stage 2 scores. That seems statistically true, but then again the total scores are discussed throughout the manuscript as lacking nuance. Also, the EYFSP is given once, correct? No change in scores at the individual level can be expected - it's a developmental health cross sectional measure. If this reviewer has missed a key point, and that they EYFSP is a longitudinally administered measure, please correct me.

In limitations, I do not follow how the authors conclude that their findings will generalize to the revised version. The revised version seems to compress data even further, limiting total score nuance even further.

Implications and future directions: does this manuscript really support continued use of this measure, or should it be revised to support educational benchmarking and interventions? I felt a firmer stance was being taken in the discussion for the total score instead of the GLD score. This reviewer appreciates the authors consideration of how else the EYFSP should be examined. How should it change, though? With it being in its teen years after development, if the authors are calling for revisions to the method of examining it, can they also call for revisions to the tool itself?

6. PLOS authors have the option to publish the peer review history of their article (what does this mean? ). If published, this will include your full peer review and any attached files.

**Do you want your identity to be public for this peer review?** For information about this choice, including consent withdrawal, please see our Privacy Policy .

Reviewer #1: No

Reviewer #2: **Yes: ** Lauren N. Girouard-Hallam

Reviewer #3: No

---

## [Author Response · Author response to Decision Letter 1]

2 Sep 2024

SEE RESPONSE TO REVIEWERS DOCUMENT

We thank all three reviewers for their reviews of the manuscript. We provide our response to each of the reviewers points below, with our responses written in blue. Where we refer to page numbers where we have made changes in the manuscript, the numbers relate to the manuscript with tracked changes.

Reviewer #1:

Thank you for the opportunity to review the manuscript, “An assessment of the teacher completed ‘Early Years Foundation Stage Profile’ as a routine measure of child developmental health”, submitted to PLOS One.

The authors report a secondary data analysis, which attempts to establish the internal consistency and structural validity of the Early Years Foundation Stage Profile (EYFS-P) in one ethnically diverse region in England (i.e., Bradford).

In principle, this study has the potential to inform research using this specific administrative data. However, several substantial issues within the current manuscript prevent its publication in its current form and underscore my ‘reject’ recommendation. These issues are outlined in detail below. Overall, I hope this feedback is helpful for future revisions of this work.

Response: We thank the reviewer for the thorough and rigorous review. We have made several changes to the manuscript based on their suggestions and feel it has strengthened it substantially.

‘Developmental health’ definition and implications for structural validity evaluation

Throughout the manuscript, the authors refer to the EYFS-P as a measure of ‘developmental health’ which they define as “a broad concept that combines a holistic understanding of physical, mental, social and emotional wellbeing”.

However, this definition is not well-aligned with the intended purpose of the EYFS-P, which also incorporates “specific areas of learning” (i.e., early academic skills). For example, as highlighted in the DfE (2024) guidance, the EYFS-P is designed to assess the “prime areas of learning (which are: communication and language; personal, social and emotional development; and physical development) and the specific areas of mathematics and literacy” (as well as Understanding the world and Expressive arts and design) (see page 6 https://assets.publishing.service.gov.uk/media/65253bc12548ca000dddf050/EYFSP_2024_handbook.pdf). Moreover, the early learning goals (ELGs) within the EYFS-P are reflective of the early years foundation stage curriculum (see Development Matters documentation) and thus may be better thought of as a curriculum-based measure.

As a consequence of this misalignment, the authors have misattributed the constructs measured within the EYFS-P. This has important implications for their structural validity evaluation. Specifically, they have used a confirmatory factor analysis approach with the a-priori assumption that the EYFS-P loads onto one factor (‘developmental health’).

However, given the design and intended purpose of the EYFS-P, there may be alternative factor structures that should also be considered, for example, a 2-factor structure (prime areas of learning; specific areas of learning) or a 7-factor structure (Communication and language; Personal, social and emotional development; Physical development; Literacy; Mathematics; Understanding the world; Expressive arts and design). As such, an exploratory factor analysis approach would be more appropriate in the current study.

Response: We appreciate the Reviewer’s point that the way in which we had defined ‘developmental health’ did not align with the intended purpose of the EYFSP, as it appeared to ignore some aspects of the curriculum that the EYFSP was based on. We revisited the reference for the term ‘developmental health’ (Keating et al., 1999) and found that they do include measures of mathematics and literacy in their definition. We have therefore amended the first sentence in our manuscript to ensure that the definition covers not only physical, social, and emotional wellbeing, but also “core educational abilities such as mathematics and literacy”. We feel that this clarification now better aligns with the original purpose of the EYFSP.

With regards to the issue around the confirmatory factor analysis (CFA) with all EYFSP items loading onto one factor, this was carried out as it is an essential analysis to accompany Item Response modelling. Item response models assume that the latent trait variable is reflected by a unidimensional continuum, thus this must be tested using CFA (explained in detail on p.14-15 of the manuscript). We do agree with the reviewer that an exploratory factor analysis of the EYFSP items would be interesting, to explore if the items do appear to relate to the areas they are intended to measure. However, since the intended purpose of our study was to validate the total score of the EYFSP (ie. not the EYFSP as a whole) – an EFA would not answer the objectives of our study.

Application of the current findings to current educational practice

Although the authors used the 2008-2012 version of the EYFS-P as the basis for their study, they make the following conclusions: “We expect that the results from this study will generalise to the revised version of the EYFSP, particularly the findings regarding internal consistency and predictive validity” and “This study supports future use of the EYFSP total score over the EYFSP GLD score for research and educational purposes”.

These conclusions are not supported and are problematic because the 2008-2012 version of the EYFS-P is significantly different from current educational practice. For example, the most recent statutory EYFS-P introduced in Sept 2021 includes several substantial changes to the ELGs. These changes include the removal of the Shape, Space, and Measure ELG, for the introduction of the Numerical Patterns ELG, as well as distinct changes to the content of a range of the ELGs across the EYFS-P. These changes will have important implications for the psychometric properties that the authors have examined and thus limit the generalisability of the current findings.

Furthermore, it is of concern that the authors do not acknowledge anywhere in their manuscript that the EYFS-P was revised again in 2021 (e.g., on page 4).

The authors should, therefore, clarify that their results are only applicable to research studies that are planning to use the 2008-2012 EYFS-P total score (page 27). Any further application (without further investigation) would be inappropriate.

Response: Thank you to the reviewer for raising concerns around the version of the EYFSP examined. We agree this is important, as the EYFSP has changed several times over the years, and agree it is crucial for the reader to understand which version is used.

We used the version of the EYFSP delivered between 2012-2021, not the version delivered between 2008-2012. We apologise for the confusion and believe this may be because we stated that we used the ‘original version’ of the EYFSP (p.6) – we have now updated this to say we investigate the ‘second version of the EYFSP’ (p.6). Based on the reviewer’s recommendation, we have also included a file (Technical Appendix File 1) which shows the second version of the EYFSP (the one we analysed), and the revised version (the one used in practice now). We have also made our statement clearer that the EYFSP was revised in 2021. We have also stated in the limitations of our discussion that we only considered data from this version of the EYFSP, and state that future research should test if the findings generalise to the newer version (p.34). We hope this alleviates the reviewers concerns around our conclusions.

Fine-grained analytical approach

Although the authors helpfully adopt a more fine-grained approach for considering the relationships between EYFS-P Communication and Socioemotional goals and SDQ difficulties (page 25), there presents a missed opportunity for adopting a similar approach for literacy-based ELGs to later literacy outcomes, and maths-based ELGs to later maths outcomes. This is despite the authors’ argument on page 6, which states “this rationale can be generalised to all seven areas of the EYFS-P”.

Response: We agree with the reviewer that it will be helpful to examine relationships between specific EYFSP goals and later outcomes. The Born in Bradford dataset facilitated an analysis of the longitudinal association between EYFSP goals and SDQ scores, and this is particularly useful as SDQ is a widely validated measure of children’s socioemotional abilities. However, data relating to all other areas of the EYFSP were not available in the Born in Bradford sample. Our discussion does highlight that ‘more research is needed to explore whether each specific goal area is associated with other measurements’, and we have now added in an explanation of why we selected the SDQ as the test case (p.35-36).

Psychometric terminology

Pages 6-7- The authors write “information about the construct validity (i.e. the extent to which a test measures what it is intended to measure)… or instance, do the ‘personal, social and emotional development’ areas have significant predictive associations with a validated measure of children’s social and emotional development?” This is incorrect, as it is an example of (predictive) criterion validity, not construct validity. I recommend that the authors familiarise themselves with resources, such as the AERA, APA, NCME Standards for Educational and Psychological Testing, which will help ensure that the terminology they use is accurate.

Response: Thank you for highlighting this mistake in our language. We agree with the Reviewer that this paragraph is describing predictive validity, and have updated the language to clarify this (p.8)

Page 8- The authors write “this [internal consistency] is an essential first step prior to investigating the structural validity of the EYFS-P” – this is not strictly true. Rather the COSMIN taxonomy and the AERA, APA, NCME Standards for Educational and Psychological Testing resources (mentioned above) suggest that structural validity comes first (i.e., how many factors are there?), followed by internal consistency (i.e., are the items interrelated?). Furthermore, the interrelations between items should be considered at both the full-item level (i.e., all EYFS-P items) and (if applicable) at the specific factor level (e.g., if the structural validity evaluation identified 2-factors, then the internal consistency also needs to be reported for these 2 factors). As such, I recommend that the reporting of the results in the current study be re-structured (and re-analysed per the suggestions mentioned).

Response: Thank you for raising this important point. We have now rearranged our analysis so that we first present the structural validity, followed by internal consistency.

We do appreciate the reviewers point around structural validity evaluation (e.g. potentially identifying >1 factor), however, in this case the EYFSP ‘total score’ measure is already being used – hence the structural validity section of our study aims to validate the total score only. We have explained this briefly in the rationale.

Other specific issues within the manuscript

Page 3- The authors write “The embedding of standardised measurement of this into educational systems varies greatly across countries”, but then go on to discuss one example (Canada) – are there other examples that could be included here?

We have now explained internationally used measures of child developmental health (the Early Child Developmental Index, and Early Development Instrument). We have also contextualised how these are embedded into educational systems – with the EDI into Canda and Australia, and the Teaching Standard Gold (TS GOLD) in the US (p.3-4).

Page 3- The authors write “Due to the educational pressures that standardised exam settings can bring, assessments completed by children’s teachers can instead offer a valuable insight” – this is somewhat true, but requires further clarification- e.g., Educational pressures are also not the only reason why teacher-based assessments may be considered optimal with very young children. Likewise, teacher-based assessments can also be ‘high-stakes’, such as those completed in Key Stage 1.

Response: Here we mean to refer to the stress that is caused to children by exam settings, and we have now clarified this. (p.5)

Page 3- The authors write “The EDI… has generally demonstrated adequate psychometric properties in terms of internal consistency (15), and predictive validity”. It would be helpful to report exact figures here, as the term ‘adequate’ is often inconsistently and/or misused across psychometric studies. The COSMIN taxonomy and the AERA, APA, NCME Standards for Educational and Psychological Testing, will provide helpful starting points for the authors in establishing the consistent use of the term ‘adequate psychometric properties’.

Response: We have added in the internal consistency, model fit, and predictive validity values for the measurement instruments that we describe (p.3-4).

Page 4- The authors write “69 ‘early learning goals’ (ELGs)”- this is not strictly true, rather the DfE refer to the structure of the 2008-2012 version of the EYFS-P as “13 assessment scales, with 9 points within each scale (‘scale point’). The 13 assessment scales are grouped into six areas of learning” (e.g., https://assets.publishing.service.gov.uk/media/5a7b875c40f0b62826a0429c/sfr28-2010.pdf and https://dera.ioe.ac.uk/id/eprint/8221/13/Early_FS_Handbook_v11_WO_LR_Redacted.pdf).

Response: We appreciate the reviewer’s comments on this point, and we have revisited the literature on the reforms to the original EYFSP. Our reading of the Tickell Review (reference #17) and the government’s publications on the changes to the EYFSP (see here: https://www.gov.uk/government/news/new-early-years-framework-published) confirms our understanding that the original 69 early learning goals were consolidated into the 17 used in the second version of the EYFSP.

Page 4- The authors write “the original version…has been used nationally and routinely for nine years” – this is also not strictly true, as the 2008-2012 version of the EYFS-P (the focus of the current study) was used for 4 years. The 2012-2021 revised version for a further 9 years, with the latest iteration made statutory in Sept 2021.

Response: This relates to the earlier point regarding which version of the EYFSP was used, we have added in clarification that we are looking at the second version of the EYFSP, not the original version.

Page 6- The authors highlight research evaluating the measurement properties of the EYFS-P, but do not include other relevant literature, e.g., Snowling et al. 2011 https://files.eric.ed.gov/fulltext/ED526910.pdf

Response: We have now included reference to this study (p.6).

Page 6- The authors highlight “there are seven individual learning areas within the EYFSP” – (this refers to the 2012-2021 version), but on page 4 the authors state “the present study investigates the original version” (which I assume to mean the 2008-2012 version), which would include 13 assessment scales (see above point)- further clarity is required here.

Response: This relates to an earlier point regarding which version of the EYFSP was used, we have added in clarification that we are looking at the second version of the EYFSP, not the original version.

It may also be helpful to create a table (perhaps expanding on table 1) that outlines each of the three versions of the EYFS-P, including which learning areas are measured, how they are measured (i.e., meets, exceeds, etc) and when they were implemented (i.e., years).

Response: We agree with the reviewer that a table is helpful for clarifying the areas of learning, thank you for the recommendation. We have included a table which outlines the second version of the EYFSP (the version that we use) and the revised version of the EYFSP (the version currently used in practice). We have not included the original version of the EYFSP, as this would contain lots of information (69 ELGs), and we do not feel that this version has implications for our findings (uploaded as Technical Appendix File 1).

Page 7- The authors write “In understanding these strengths and weaknesses, a child could then be

---

## [Decision Letter · Decision Letter 1]

1 Dec 2024

PONE-D-24-14293R1An assessment of the teacher completed ‘Early Years Foundation Stage Profile’ as a routine measure of child developmental healthPLOS ONE

Dear Dr. Mooney,

Thank you for submitting your manuscript to PLOS ONE. After careful consideration, we feel that it has merit but does not fully meet PLOS ONE’s publication criteria as it currently stands. Therefore, we invite you to submit a revised version of the manuscript that addresses the points raised during the review process.

Dear Dr. Mooney, 

Thank you for your patience in the review process and the detailed response to each of the original reviewers' critiques.  These revisions have greatly improved your manuscript.  Please thoroughly respond to each of the new reviewers' critiques and resubmit your manuscript.  Please use titles of the reviewer number, spaces, and numbers to itemize your changes so that it is easier to follow and read.  Please contact our team with any questions.  

We look forward to receiving your revised manuscript.

Kind regards,

Nhu N. Tran

Academic Editor

PLOS ONE

Journal Requirements:

Reviewers' comments:

Reviewer's Responses to Questions

**Comments to the Author**

1. If the authors have adequately addressed your comments raised in a previous round of review and you feel that this manuscript is now acceptable for publication, you may indicate that here to bypass the “Comments to the Author” section, enter your conflict of interest statement in the “Confidential to Editor” section, and submit your "Accept" recommendation.

Reviewer #2: All comments have been addressed

Reviewer #4: (No Response)

Reviewer #5: All comments have been addressed

2. Is the manuscript technically sound, and do the data support the conclusions?

Reviewer #2: (No Response)

Reviewer #4: Yes

Reviewer #5: Yes

3. Has the statistical analysis been performed appropriately and rigorously?

Reviewer #2: (No Response)

Reviewer #4: Yes

Reviewer #5: Yes

4. Have the authors made all data underlying the findings in their manuscript fully available?

Reviewer #2: (No Response)

Reviewer #4: Yes

Reviewer #5: Yes

5. Is the manuscript presented in an intelligible fashion and written in standard English?

Reviewer #2: (No Response)

Reviewer #4: No

Reviewer #5: No

6. Review Comments to the Author

Reviewer #2: (No Response)

Reviewer #4: Again, nice work but needs some copy editing and then the results and discussion need to be redone to be more traditional

See the note for more information

Reviewer #5: This article examined the internal consistency , structural validity and predictive validity of the EYFSP.It is a longitudinal study with ethical clearance obtained from concerned authority and written consent taken.

Clarifications required- in page 8 this sentence has to be framed differently " and it was not reported how EYFSP subscale scores were calculated for this study" to be framed as above mentioned study.( Reference no 28)

Page 11 " routine educational data was collected....from local authority every year the child attended school" there is no mention at what age the first educational data was collected. Next additional data was collected at 7-10 years but In the discussion of third aim of predicting academic outcome the authors mentioned outcome of 10-11 years.

Authors to kindly clarify " overlap in total scores and variability in scores who does and does not meet GLD" accompanied table of overlap items would help to clarify and assist teachers and researchers.

The research has mentioned limitations and future directions.

7. PLOS authors have the option to publish the peer review history of their article (what does this mean? ). If published, this will include your full peer review and any attached files.

**Do you want your identity to be public for this peer review?** For information about this choice, including consent withdrawal, please see our Privacy Policy .

Reviewer #2: No

Reviewer #4: **Yes: ** M. Diane Clark

Reviewer #5: **Yes: ** Shabina Ahmed

---

## [Author Response · Author response to Decision Letter 2]

3 Jan 2025

SEE ATTACHED WORD DOCUMENT.

Response to reviewers, 03/01/2024

We thank both Reviewer’s for their consideration of our study. Their review has helped to strengthen the manuscript. We provide responses to each Reviewer’s comments below. The page numbers we refer to relate to the manuscript with tracked changes.

Comments to the Author

Reviewer #5:

This article examined the internal consistency , structural validity and predictive validity of the EYFSP. It is a longitudinal study with ethical clearance obtained from concerned authority and written consent taken.

Clarifications required- in page 8 this sentence has to be framed differently " and it was not reported how EYFSP subscale scores were calculated for this study" to be framed as above mentioned study. ( Reference no 28)

Response: Thank you for noting this. We have reworded this sentence to make it clear that we are referencing the aforementioned study (see page 8).

Page 11 " routine educational data was collected.... from local authority every year the child attended school" there is no mention at what age the first educational data was collected. Next additional data was collected at 7-10 years but In the discussion of third aim of predicting academic outcome the authors mentioned outcome of 10-11 years.

Response: We have now explained that “educational outcomes were obtained from the Local Authority every year that the child attends school, starting at age 4 (Reception year)” (page 11).

Thank you for raising that it is not clear which age our outcomes were collected at. Outcomes for RQ3 were collected at age 10-11 through the routine educational data, and outcomes for RQ4-5 were collected in bespoke data collection by Born in Bradford at ages 7-10 (see page 17 for an explanation).

We have made amendments to the language in our discussion to make it clear when the outcomes were collected. In the discussion we refer to ‘outcomes at ages 10-11 in Maths, Reading, and Grammar…”. We then explain that we ‘explore the predictive validity of the EYFSP total score and the EYFSP-CS subscales for children’s behavioural, social, and emotional difficulties”. We have now amended this sentence to explain that these were collected “at ages 7-10”, which aligns with the additional data collection explained in the methods (see page 31).

Authors to kindly clarify " overlap in total scores and variability in scores who does and does not meet GLD" accompanied table of overlap items would help to clarify and assist teachers and researchers.

The research has mentioned limitations and future directions.

Response: Thank you for raising this and suggesting a table to clarify the overlap in items. In response to the reviewer’s suggestion, we did explore if it would be possible to include a table here, but we would have to suppress numbers within some cells due to the risk of reidentification. We instead refer to the figure as this demonstrates the overlap in total scores between those who do and do not achieve a GLD, without displaying the actual frequencies and risking reidentification of participants. We hope that this addresses the Reviewer’s concern (see figure below).

Figure 2. Kernel density distributions of EYFSP total score for those who do not achieve a GLD (in blue) and do achieve a GLD (in orange) (n=10,589).

Reviewer #4:

Again, nice work but needs some copy editing and then the results and discussion need to be redone to be more traditional

See the note for more information

Response: We thank the Reviewer for their review of our study. Please see our response below to Reviewer #4.

Attached note from Reviewer #4

Early Years Foundation Stage Profile (EYESP)

Interesting article that I enjoyed reviewing ---thanks.

Please look up the manuscript guidelines for Plos One

You need to clean up spacing and when you indent and when you do not indent the first

paragraph after a heading

Only three levels of headings are permitted

The references need to be reformatted.

Response: Thank you for noting this. We have: (1) cleaned up our spacing throughout, ensuring we are using a spacing of 2.0 for all lines, (2) ensured that we do not indent the first paragraph after each heading, (3) amended our heading levels so that we use a maximum of three levels of headings throughout, and (4) checked the formatting of referencing throughout.

In response to the Reviewer’s earlier suggestion that the manuscript requires some copy-editing, we have re-read the manuscript and made minor changes throughout to improve its readability. Thank you.

Your discussion is really results that should be more clearly related to your hypothesis

Then in the discussion it should bring in the lit review with your findings

Response: In addition to this comment, the Reviewer suggests that the results and discussion need to be redone to be more traditional. In response to these comments, we have examined the overlap between the literature covered in our introduction and discussion. Due to there being no previous psychometric research on the EYFSP total score, there is a lack of literature to bring in for Research Questions 1-2. For Research Questions 3-5 regarding the predictive validity, our discussion already mentions the prior literature (see pages 30-31). However, we have made the following changes to ensure our discussion covers the literature in our introduction as much as possible (with changes highlighted in underlined italics):

• Included the full explanation of child developmental health as mentioned in our introduction: “The construct of developmental health encompasses a holistic understanding of children’s physical, mental, social, and emotional wellbeing, combined with core educational abilities such as mathematics and literacy (1)” (see page 29)

• Included reference to studies which describe the methods we have used: “The first aim was to investigate the internal consistency the EYFSP items (44,47)” (see page 29)

• Included a reference to two studies from our introduction into our discussion section: “Although both the GLD and now the total score have been shown to predict future outcomes (34,35),” (see page 32)

• Included literature relating to the psychometric properties of other measures of child developmental health on page 34: “We do not suggest the use of a different measure of child developmental health over the EYFSP. Whilst other measures of child developmental health have undergone further psychometric analyses, they too demonstrate some variable psychometric properties (e.g. variable model fit values have been demonstrated in both the EDI (17) and TS-GOLD (19))” (see page 34)

To further address the Reviewer’s comments, we have confirmed that our current format adheres to the PLOS ONE submission guidelines and style recommendations (https://journals.plos.org/plosone/s/submission-guidelines). The discussion section thoroughly addresses each research question and integrates relevant literature wherever possible.

We hope that the changes made outlined here, and the clarifications we have provided on the PLOS ONE guidelines, will effectively address the Reviewer's comments. Whilst we understand that the Reviewer may have intended more significant revisions, we would kindly request further information on the specific changes to ensure that results and discussion sections are traditional, if the Reviewer still deems this necessary for publication of our study.

Small issues

In the marked up version ( there are no track changes) page 25 I found this (45,48). (47)

Not sure why the 47 is outside the period

Response: Thank you for noting this error, we have now amended this and removed the 47 outside of the period (see page 9).

A change in 6 in the EYFSP total score results in a change of -1.22 in SDQ, and a change in 6 in EYFSP-

Seems it should be of 6 in the EYESP (in both places)

On page 44

Response: Thank you. We have now amended this to instead say ‘a change of’ (see page 28).

Despite this, this study

----avoid the double this page 50

Response: Thank you, we have no amended this to instead say ‘despite this, the present study’ (see page 34).

---

## [Editor Report · Decision Letter 2]

14 Jan 2025

PONE-D-24-14293R2An assessment of the teacher completed ‘Early Years Foundation Stage Profile’ as a routine measure of child developmental health

PLOS ONE

Dear Dr. Mooney,

Thank you for submitting your manuscript to PLOS ONE. After careful consideration, we feel that it has merit but does not fully meet PLOS ONE’s publication criteria as it currently stands. Therefore, we invite you to submit a revised version of the manuscript that addresses the points raised during the review process.

**Thank you for revising and resubmitting your manuscript!  The revisions have strengthened the manuscript.  Please consider the revisions detailed below. **

**Discussion:**

**This section continues to lack detail and robustness. Please revise this section to review how your study’s results relate to the other existing literature. It should not be a repeat of the rationale from the introduction section. If studies that have examined your topics do not exist, then explicitly write that, however, other studies may have had similar study designs, etc. If that is the case, those studies results should be compared to your findings.**

**Please copy paste the changes to your manuscript into the Rebuttal/response letter so that it is easier for the reviewers to access.  **

We look forward to receiving your revised manuscript.

Kind regards,

Nhu N. Tran

Academic Editor

PLOS ONE
---

## [Author Response · Author response to Decision Letter 3]

21 Jan 2025

Reviewer comment:

1. Discussion:

This section continues to lack detail and robustness. Please revise this section to review how your study’s results relate to the other existing literature. It should not be a repeat of the rationale from the introduction section. If studies that have examined your topics do not exist, then explicitly write that, however, other studies may have had similar study designs, etc. If that is the case, those studies results should be compared to your findings.

Please copy paste the changes to your manuscript into the Rebuttal/response letter so that it is easier for the reviewers to access.

Thank you for helping us revise our discussion regarding how our results relate to the other existing literature on comparable measures of child developmental health. We have made the following changes:

1. We have added two paragraphs explaining previous studies which have used IRT to evaluate comparable measures of child developmental health, and related the EYFSP’s internal consistency and structural validity to comparable measures of child developmental health in previous literature [pages 29-30, version with no tracked changes]:

“Although IRT has been used to examine measures of, for example, emotional dysregulation (76) and neuropsychological capabilities (77) of young children, we have not been able to locate any previous studies using IRT on a comparable measure of broad child developmental health. The closest study is an investigation of the ‘Denver Developmental Screening Test (DDST)’, though the aim of this study was to apply IRT to develop a scoring method to estimate ‘ability age’ of individual children using the DDST (78). Hence, IRT has been underused in the context of examining the structural validity of child developmental health measures. Our study is therefore first to investigate the EYFSP’s internal consistency and structural validity using IRT, and one of the first to apply this method to early child developmental health.

Nonetheless, the internal consistency and structural validity of comparable child developmental health measures has been investigated using other psychometric analysis methods, namely the EDI and TS GOLD. The EYFSP has now demonstrated similar adequate internal consistency as the EDI (17), and demonstrates model fit similar to both the EDI and TS GOLD, which have also shown poor model fit (17,19). Model fit refers to the ability of a model to reproduce the data, with poor model fit indicating that relationships between variables may be incorrectly specified in the applied statistical model (57). Given that the EYFSP, EDI, and TS-GOLD have all demonstrated poor model fit, this may indicate a general challenge with using teacher reported measures of overall child developmental health. As this is such a broad construct, perhaps it is challenging to assess in one holistic measure. Indeed, a systematic review of parent-reported measures of child social and emotional wellbeing/behavior found such measures to have structural validity (79).

2. Following the above paragraphs, we have kept our previous discussion section which discusses the potential reasons for poor model fit. We have then added the following paragraph [page 31]:

Evidence from comparable measures of child developmental health comes from numerous psychometric studies conducted in several different countries over several years (17,19). Hence, to be able to thoroughly compare the EYFSP to these similar measures, more research on the measurement properties of the EYFSP is needed (see implications and future directions section).

3. Following the above paragraph, we then discuss the results from the predictive validity analysis. This section already considered the previous studies which have investigated the predictive validity of the EYFSP. To further situate our findings existing literature we have added the following paragraph [page 32-33]:

“There is only one other study that has reported the predictive validity of this version of the EYFSP subscales; it found that EYFSP scores relating to literacy and physical development also predicted children’s behavioral, social, and emotional difficulties (27). In comparison to other measures of child developmental health, the EDI has demonstrated variable predictive validity between the language and cognitive development domain scores and the Peabody Picture Vocabulary Test (17). The TS GOLD has been found to be associated with children’s assessments throughout the school year (80), and has been found to have variable concurrent validity with the Bracken School Readiness Scale (19). The EYFSP therefore has demonstrated adequate predictive validity in comparison to other measures of child developmental health, however, there are substantially fewer studies regarding the EYFSP.”

4. Our previous discussion then had a ‘limitations’ section, followed by an ‘implications and future directions’ section. To improve the flow of the new discussion, and to situate the ‘implications and future directions’ within the new literature we have now discussed, we have rearranged the remainder of the discussion so that the sections are ordered as follows:

Implications and future directions

Limitations

Conclusions

5. Related to the above point, we have moved a paragraph to the start of our implications and future directions section to compare the EYFSP to other comparable measures as now discussed earlier [page 34]:

Although this study has highlighted some limitations of using the EYFSP, we do not suggest the use of a different measure of child developmental health over the EYFSP. Whilst other measures of child developmental health have undergone further psychometric analyses, they too demonstrate some variable psychometric properties (e.g. variable model fit values have been demonstrated in both the EDI (17) and TS-GOLD (19)). Replacement of the EYFSP would be a significant overhaul to current educational practice and should be avoided if possible. Hence, once the measurement properties of the EYFSP are investigated as described above, the EYFSP could be used with more confidence than it currently is. Or, if the measurement properties of the EYFSP are found to be significantly lacking, a significant programme of development should be undertaken to develop it further, or replace it with an already validated instrument of child developmental health.

6. Finally, we have made minor copy edits throughout to improve the flow of the discussion. This includes the removal of a paragraph at the start of the introduction that reiterated the rationale for our study to ensure that our discussion is dedicated to discussing the findings in relation to the wider literature:

[Now removed] Embedding routine measurement of children’s developmental health into educational systems is crucial to provide support to those who need it (10,11), and potentially close inequalities in educational outcomes (9). In England and Wales, the EYFSP with 17 goals has been routinely completed by teachers for all children attending school for nearly ten years. Due to the potential use of the EYFSP ‘total score’ for both research studies and applied educational settings, we investigated whether it is fit for purpose as an overall summary of child developmental health [Now removed]

---

## [Editor Report · Decision Letter 3]

28 Jan 2025

An assessment of the teacher completed ‘Early Years Foundation Stage Profile’ as a routine measure of child developmental health

PONE-D-24-14293R3

Dear Dr. Mooney,

We’re pleased to inform you that your manuscript has been judged scientifically suitable for publication and will be formally accepted for publication once it meets all outstanding technical requirements.

Kind regards,

Nhu N. Tran

Academic Editor

PLOS ONE
---

## [Editor Report · Acceptance letter]

PONE-D-24-14293R3

PLOS ONE

Dear Dr. Mooney,

I'm pleased to inform you that your manuscript has been deemed suitable for publication in PLOS ONE. Congratulations! Your manuscript is now being handed over to our production team.

Kind regards,

on behalf of

Dr. Nhu N. Tran

Academic Editor

PLOS ONE